# Investigating the influence of changing ice surfaces on gravity wave formation impacting glacier boundary-layer flow with large-eddy simulations

Brigitta Goger[1,2], Ivana Stiperski[2], Matthis Ouy[2], and Lindsey Nicholson[2]

[1]Center for Climate Systems Modeling, ETH Zurich, Zurich, Switzerland
[2]Department of Atmospheric and Cryospheric Sciences, Universtität Innsbruck, Innsbruck, Austria

**Correspondence:** Brigitta Goger (brigitta.goger@c2sm.ethz.ch)

**Abstract.** Mountain glaciers are located in highly complex terrain and their local microclimate is influenced by mountain boundary layer processes and dynamically-induced gravity waves. Previous observations from turbulence flux towers, as well as large-eddy simulations, over the Hintereisferner (HEF) glacier in the Austrian Alps have shown that down-glacier winds are often disturbed by cross-glacier flow from the North-West associated with gravity waves. In this work, we explore how changing the ice surface coverage upstream of HEF influence this gravity wave formation and intensity and the feedback this has on boundary layer flow over HEF. In semi-idealized large-eddy simulations, we explore the impact of changing surface properties on HEF's microclimate by removing the upstream glaciers only (NO_UP) and removing all ice surfaces (NO_GL). Simulations suggest that removing the upstream glaciers (which causes a change of boundary layer stratification from stable to unstable) leads to a weaker gravity wave that breaks earlier than in the reference simulation, resulting in enhanced turbulent mixing over HEF. As a consequence, this leads to higher temperatures over HEF tongue. Removing all glaciers results - as expected - in higher temperatures of up to 5 K over the missing ice surfaces, while the gravity wave pattern is similar as in the NO_UP simulation, indicating that the upstream boundary layer exerts dominant control over downstream response in such highly dynamic conditions. Furthermore, the results show that the upstream glaciers have a stabilizing effect on the boundary layer, impacting gravity wave formation and downslope windstorm intensity and their feedback on the flow structure in valleys downstream. This case study shows that a single glacier tongue is not isolated from its environment under strong synoptic forcing and that surrounding glaciers and local topography have to be taken into account when studying atmosphere-cryosphere exchange processes.

## 1 Introduction

The Earth's mountainous regions are important sources for global freshwater, and are strongly affected by climate change and elevation-dependent warming (Hock et al., 2022; Byrne et al., 2024; Pepin et al., 2022). In recent years, the European mountain cryosphere (permafrost, glaciers, snow) has undergone significant change (Beniston et al., 2018) with decreasing overall snow depths (Matiu et al., 2021) and rapid and accelerating glacier recession (e.g., Voordendag et al., 2023; Cremona et al., 2023; Rounce et al., 2023). While there is a clear scientific consensus on the large-scale patterns of glacier recession, there is still a

knowledge gap on how glacier shrinkage influences the mountain boundary layer (MoBL, Lehner and Rotach, 2018) and how
that in turn will influence subsequent glacier melt (Beniston et al., 2018).

The MoBL represents a complex multiscale interaction between the surface, complex heterogeneous topography and the atmosphere aloft, with MoBL processes occurring on timescales of one hour to several hours (Rotach and Zardi, 2007; Lehner, 2024; Pfister et al., 2024). One of the major features of the MoBL are thermally-induced circulations at multiple scales, from slope flows to valley winds up to Alpine pumping (Zardi and Whiteman, 2013; Goger and Dipankar, 2024) which together
can influence the exchange of heat, mass, and momentum between the surface and the free atmosphere. European glaciers are located in mountainous terrain, and although they develop their own microclimate with persistent katabatic down-glacier flows, they are also affected by valley winds and/or the larger-scale synoptic flows (Oerlemans, 2010; Potter et al., 2018). Under a warming climate, recent observations and modelling over the Haut glacier d'Arolla in the Swiss Alps suggest that with a decreasing glacier ice area the thermally-induced up-valley flow progressively dominates the glacier microclimate and
ultimately might contribute to increased melting (Shaw et al., 2023, 2024). Conway et al. (2021) suggest that local glacier boundary-layer flow is also influenced by local breeze systems induced by a adjacent, larger icefields ("Icefield breezes"), and that a single glacier tongue is not isolated from its environment.

Besides thermally-induced flows, larger-scale synoptic forcing has a non-negligible impact on local glacier boundary layers (e.g., Litt et al., 2017; Mott et al., 2020). When stratified flow is displaced vertically by topography, it is forced to leave
its hydrostatic equilibrium, leading to the formation of gravity waves, more specifically lee waves (e.g., if an inversion or a decrease of Scorer parameter with height is present, Scorer, 1949), where the lee-wave amplitude depends on the topography height, shape, and the upstream profiles of vertical wind shear and atmospheric stability (Jackson et al., 2013). Gravity waves are common over the Alps (Jiang and Doyle, 2004), and large-amplitude gravity waves can lead to supercritical flow in the lee of mountains resulting in downslope windstorms (e.g., foehn winds, Gohm and Mayr, 2004). The supercritical flow can form
due to wave breaking aloft (visualized as overturning isentropes and associated with severe turbulence) or along an inversion, can plunge into valleys, interacting with the local boundary layer and modifying its spatio-temporal structure (Jiang et al., 2006; Adler and Kalthoff, 2016; Kalthoff et al., 2020; Haid et al., 2022), and under appropriate conditions can lead to boundary-layer separation (French et al., 2015; Grubišić et al., 2015) and turbulent flow reversal at the surface, associated with the formation of atmospheric rotors (Grubišić et al., 2008; Strauss et al., 2015; Vosper et al., 2018).
Because melting glaciers exhibit a constant surface temperature of 0°C, and are comparatively smoother than ice-free mountainous areas, glaciers can themselves influence the very formation of gravity waves. In a study of foehn flow over the Larsen C ice shelf, Antarctica, Turton et al. (2018) hypothesize that upstream ice surfaces influence the isentrope downdraw in downslope windstorms. A numerical sensitivity study over Hofsjökull ice cap, Iceland, suggests that downslope windstorms are stronger due to the stabilizing effect of the ice surfaces on gravity waves, while removing the icecap from the simulation domain led
to weaker gravity waves (Jonassen et al., 2014), where the authors attribute this pattern change mostly to a different surface roughness with a minimal effect of temperature change.

Many of the aforementioned phenomena were observed during the Hintereisferner Experiment (HEFEX, Mott et al., 2020), a measurement campaign at the Hintereisferner (HEF) glacier, Austrian Alps. Along- and across-glacier transects of eddy-

covariance (EC) stations allowed the analysis of spatial heat advection patterns. The distinct katabatic down-glacier flow was often disturbed by lateral (cross-glacier) flow in 20% of the data. Within the HEFEX observational setup, the reason for these disturbances could not be immediately identified and one of the major questions was the origin of this cross-glacier flow. A large-eddy simulation (LES) setup was used to investigate the nature and source of these cross-glacier disturbances (Goger et al., 2022). With a horizontal mesh size of 48 m, the model could simulate mesoscale wind patterns on the glacier for both summer and winter successfully, but small-scale glacier boundary layer features were not resolved accurately (e.g., the katabatic down-glacier flow, Goger et al., 2022; Voordendag et al., 2024).

Under North-Westerly synoptic flow, the simulation showed that a gravity wave forms upstream of HEF over the surrounding, smaller glaciers. In agreement with the observations, the LES showed that this gravity wave was responsible for the erosion of the glacier boundary layer over the glacier tongue. However, the role of the upstream glaciers on gravity wave formation influencing these cross-glacier "disturbances" remains unclear. Given the influence of boundary layer type on the gravity wave response, we hypothesize that the upstream glaciers have a crucial influence on the flow on HEF. In the follow-on study presented here, we therefore expand on the Hintereisferner case study under North-Westerly flow from Goger et al. (2022) by replacing upstream glacier surfaces with bare rock in a semi-idealized setup to address the following research questions:

– What is the impact of removing upstream glacier surfaces on gravity wave formation, breaking and associated downstream flow structure on the HEF glacier?

– How are surface exchange and temperature patterns on the glacier influenced by the modified flow structure after removing upstream glacier surfaces?

The paper is organized as follows: In Section 2, we describe the area of interest, our model set-up, and the analyses presented. The results are presented in two sections: Section 3 describes the impact of the changing ice surfaces on the wind structure and upstream profiles, followed by Section 4, where we investigate the impact of the missing upstream glaciers on the sensible heat fluxes, advection patters, heat budget, and 2 m temperatures on the (remaining) glacier, before we discuss the results and conclude.

## 2 Methods

### 2.1 The Hintereisferner (HEF) glacier

Hintereisferner (HEF) is a large valley glacier in the Ötztal Alps, Austria. In 2018 (the year of our case study), HEF was around 6.3 km long, descending from its highest point, Weißkugel (3738 m asl), to 2460 m asl, where the glacier tongue terminates. The glacier has been subject to continuous mass balance monitoring and meteorological observations for more than 70 years (Obleitner, 1994; Strasser et al., 2018) and is one of the benchmark glaciers of the World Glacier Monitoring Network (WGMS, 2017). The glacier is, as the rest of the European mountain cryosphere (Beniston et al., 2018), affected by persistent and accelerating mass loss since the 1980's. While HEF loses around 1 m ice thickness per year over the last 20 years (Piermattei

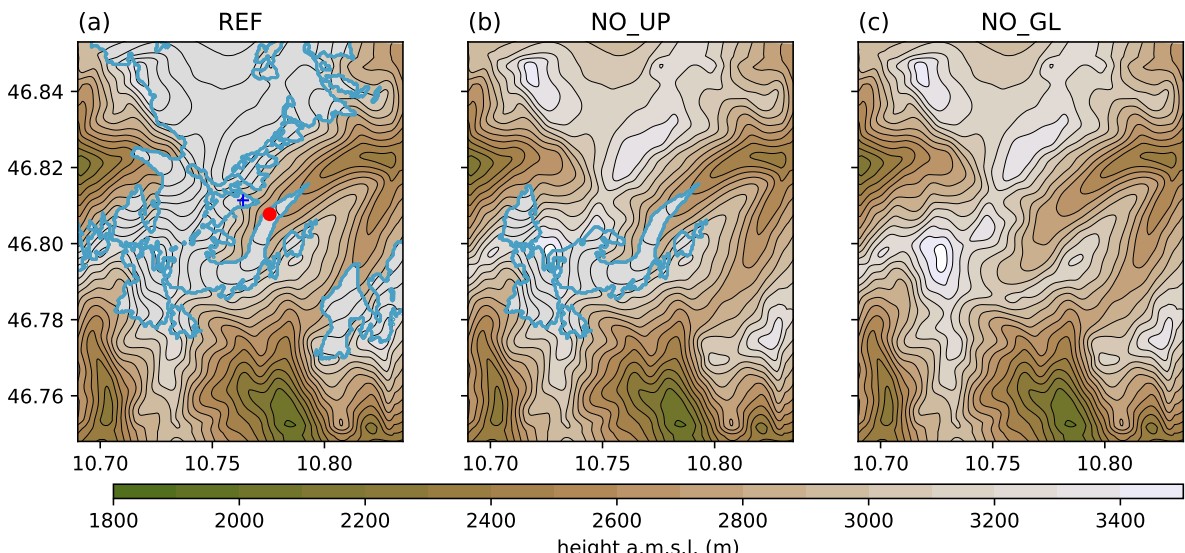

**Figure 1.** Overview of the model topography of the innermost domain (colors and contour lines) and the glacier outlines (light blue) for (a) the reference run with realistic glacier outlines (REF), (b) no upstream glaciers (NO_UP), and (c) without glaciers (NO_GL). The black lines show the cross-section lines used in the analysis, the red dot shows the location "HEF tongue", and the red star shows the location "upstream" used in the discussion of the results.

et al., 2024), extreme mass loss has been observed in some recent years (-3319 kg m$^{-2}$ in 2022, 3.2 times higher than the long-term mean for 1991-2020, Voordendag et al., 2023). The HEFEX campaign took place on the glacier in the summer of 2018 (Mott et al., 2020), revealing persistent katabatic down-glacier flow but also frequent intrusions from the North-West (20% of observation data), leading to an erosion of the glacier boundary layer. These disturbances were investigated with numerical simulations by Goger et al. (2022), revealing gravity waves over the North-Western ridge to be the major mechanism behind the intrusions into the katabatic flow.

| Land-use category | Albedo (%) | Moisture availability (%) | Emissivity (% at 9 $\mu$m) | $z_0$ (cm) | Thermal inertia (1 W m$^{-2}$ k$^{-1}$ s$^{1/2}$) |
|---|---|---|---|---|---|
| Snow or ice | 41.5 | 95 | 96.1 | 5 | 418 |
| Bare rock | 16.9 | 2 | 96.5 | 10 | 2948 |

**Table 1.** Surface parameters from the CORINE dataset for the two land-use categories 'snow or ice' and 'bare rock' for the summer season after Pineda et al. (2004).

## 2.2 Numerical model

We employ the Weather Research and Forecasting (WRF) model version 4.1 (Skamarock et al., 2019) for our numerical study. Most of the model set-up is as in Goger et al. (2022), therefore we only repeat the most relevant information for our present study. We use a nested set-up consisting of four domains, where the outermost domain spans Europe with $\Delta x$=6 km and receives ERA5 reanalyses (Hersbach et al., 2020) as boundary and initial conditions. We subsequently nest down over $\Delta x$=1 km, and $\Delta x$=240 m to the innermost domain at $\Delta x$=48 m (Goger et al., 2022, their Figure 1b). The lowest model level in the innermost domain is at 7 m, resulting in the lowest model half-level height of $z = 3.5$ m. We use the Thompson microphysics (Thompson et al., 2008), the MM5 revised surface layer scheme (Jiménez et al., 2012), and the RRTMG two-stream radiation scheme (Iacono et al., 2008) with topographic shading for all domains. We switch off the boundary layer parameterization in the two innermost domains and employ the turbulence closure after Deardorff (1980). Since the boundary-layer flow is of turbulent nature, we utilize the online averaging module `WRF LES diagnostics` by Umek (2020) and create 15-minute averages of selected model variables.

The reference simulation (REF) is a real-case simulation of the glacier boundary layer from August 17, 2018 (Fig. 1a), where the synoptic flow direction was mostly North-Westerly, resulting in sustained disturbance of the glacier boundary layer (Goger et al., 2022, their NW day case study). For this study we conduct two additional sensitivity simulations of domain 3 ($\Delta x$=240 m) and domain 4 ($\Delta x$=48 m) with changes in glacier ice surfaces: For the first sensitivity run, we replace the upstream glacierised area North of HEF (NO_UP, Fig. 1b), with the land-use category of the surroundings, namely 'bare rock', while the topography remains the same. For the second sensitivity run, we replace all glacier surfaces in the domain, including HEF, with the 'bare rock' land category (NO_GL, Fig. 1c). Table 1 shows the differences in surface parameters between the two land-use categories according to Pineda et al. (2004).

At this point, we want to mention that the two sensitivity runs do not represent realistic glacier surfaces under future climate projections (Zekollari et al., 2019), because these projections predict a continuous shrinkage of all ice surfaces instead of removing the entire upstream glaciers as in our NO_UP run. Furthermore, real world glacier recession progresses from lower to higher elevations, while our case study involves the unrealistic removal of high-elevation ice surrounding HEF. However, the aim of this study is not to investigate the glacier boundary layer development under future climate scenarios, but rather to isolate and explore the role of the upstream glacier land cover on the local boundary layer over HEF and the associated surface exchange. Therefore, we consider the NO_UP and NO_GL runs as "semi-idealized" simulations.

All simulations were initialized on Aug 17 at 03:00 UTC and ran for 18 hours. All further time information in this publication refers to UTC, so we will omit 'UTC' at all further occurrences. The first three hours of simulation time are considered as model spin-up (while the land-use remains constant for the entire simulation time), because our phenomena of interest (gravity waves, boundary-layer processes) have time scales of an hour or less. We investigate the time period from 06:00 until 12:00, because REF delivered most reliable results in comparison with observations from the HEFEX campaign during this time period (Goger et al., 2022). REF starts to deviate from the observations after 12:00 (Goger et al., 2022, their Fig. 2) due to the well-known scale separation problem in LES (Schemann et al., 2020; Umek et al., 2021). This time period is relatively

short, but since observations suggest that the situation over the glacier does not change drastically from 12:00 until sunset, an extended analysis does not bring new insights. Since we analyze a situation dominated by dynamically-induced processes, we assume that thermal effects, such as the thermally-induced valley flow circulation, are secondary. Furthermore, we only show the output of domain 4 for our analysis, and any mentioned numerical data will stem from this domain at $\Delta x$=48 m.

At the current horizontal grid spacing ($\Delta x$=48 m) not all scales in the LES are resolved equally well. As for all real-case
LES, the scale separation problem (isolating the smaller from the larger scales) is inherent (Schemann et al., 2020). This leads to a 'better' representation of the larger scales (e.g., the dynamically-induced gravity wave), which is also evident in the NW day simulation of Goger et al. (2022), where they noted a too strong erosion of the glacier boundary layer by the gravity wave. Furthermore, we cannot expect that the small-scale stable boundary layer over HEF is resolved accordingly (both in the vertical and horizontal), because according to Cuxart (2015), a horizontal grid spacing of less than 10 m is necessary to simulate stable
boundary layers in a realistic way. Still, since we focus in this study mostly on dynamically-driven processes, we think that we can provide important information on the impact of gravity wave formation on the glacier boundary layer flow development.

## 2.3 Analyses performed

### 2.3.1 Cross-sections

The atmospheric structures of the different simulations are presented as vertical slices along the transects shown in Fig. 1, and
145 the vertical profiles of the atmosphere are also presented for the two marked locations at the edge of the upstream glaciers and over the HEF glacier tongue.

### 2.3.2 Scorer parameter

To explore how the upstream profiles influence gravity wave formation, we show vertical profiles of potential temperature, wind speed, wind direction, and the Scorer parameter (Scorer, 1949) from HEF tongue and from a point located over the
150 upstream glaciers ("upstream") in Fig. 4. The Scorer parameter is used to check whether atmospheric conditions are favorable for gravity wave formation and is defined as

$$l^2(z) = \frac{N^2}{U^2} - \left(\frac{\partial^2 U}{\partial z^2}\right)/U \tag{1}$$

dependent on the height $z$, $U = U(z)$ is the vertical profile of the horizontal wind, and $N = N(z)$ is the Brunt-Väisälä frequency

$$N(z) = \sqrt{\frac{g}{\theta}\frac{\partial\theta}{\partial z}}, \tag{2}$$

with $\theta$ as the potential temperature and $g$ the acceleration due to gravity. When $l^2(z)$ decreases or changes strongly with height (e.g. existence of an inversion or increasing wind speed with height), conditions are favorable for the formation of trapped lee waves.

### 2.3.3 Up-valley wind index (UWI) and flow channelling

To disentangle the dynamical mechanisms related to the wind direction differences with the HEF valley and aloft due to changes in synoptic flow, we first compute the "up-valley wind index" (UWI) based on the wind direction, developed by Shaw et al. (2023) as follows:

$$UWI = \cos(\frac{|wdir - \phi|\pi}{180}) \tag{3}$$

where $wdir$ is the wind direction at 10 m above ground at HEF tongue (while Shaw et al. (2023) used observations from 2 m above ground), and $\phi$ is the orientation of the glacier valley (in our case $45°$). When $UWI = 1$, the flow is exactly up-glacier, while $UWI \approx 0.5$ indicates cross-glacier flow, and $UWI \approx 0$ implies down-glacier flow. While the $UWI$ gives an overview of the wind direction over the glacier it does not give information on the kind of atmospheric forcing leading to the wind direction shift.

Whiteman and Doran (1993) defined four scenarios of interactions between the (synoptic) flow aloft and the flow within a valley: thermally-driven, downward momentum transport by gravity waves, forced channeling, and pressure-driven channeling. To learn about which forcing is responsible for the wind direction in the glacier valley in our simulations, the wind direction of the synoptic flow aloft (in our case the upstream location) and the wind direction of the valley flow can be compared via a scatter plot. The resulting pattern shows whether the flow falls into one of the four categories of Whiteman and Doran (1993), their Figure 1.

### 2.3.4 Heat advection patterns and heat budget

Observations and numerical simulations agree that the local flow patterns over HEF strongly affect the heat transport and advection processes over the glacier tongue (Mott et al., 2020; Goger et al., 2022; Haugeneder et al., 2024). Therefore, in order to assess the sensitivity of heat transport to the changed glacier surface cover we calculate the horizontal temperature advection as in Goger et al. (2022):

$$T_{ADV} = -\overline{U}\frac{\Delta\theta}{\Delta s}, \tag{4}$$

where $\Delta s$ is the distance between the stations, $\Delta\theta$ is the temperature difference, and $\overline{U}$ is the average horizontal wind speed. In order to consider the total vertical heat budget, we calculate the temperature tendency equation (Wyngaard, 2010):

$$\frac{\partial\theta}{\partial t} = \underbrace{-u\frac{\partial\theta}{\partial x} - v\frac{\partial\theta}{\partial y} - w\frac{\partial\theta}{\partial z}}_{Adv} - \underbrace{\frac{\partial\overline{w'\theta'}}{\partial z}}_{vHFD} \tag{5}$$

with temperature advection with the mean wind in all three directions ($u$ is zonal, $v$ meridional, and $w$ vertical) and the vertical heat flux divergence. Here the turbulent sensible heat flux $\overline{w'\theta'}$ contains both the resolved and subgrid scale (SGS) contributions. We neglect the radiative flux divergence, because we consider it small during daytime. Horizontal averages are taken over $\Delta x = 48$ m.

### 2.3.5 Resulting temperature on the glacier

Finally, the impact of changing the ice surface extent on the simulated 2 m temperature is assessed. We choose the diagnostic 2 m temperature instead of the skin temperature, because the skin temperature is constant at 0°C over the melting glacier. The 2 m temperature calulation is based on Jiménez et al. (2012), their equation 4:

$$T_{2m} = T_g + (T_a - T_g)\frac{\ln\left(\frac{2}{z_0}\right) - \psi_h\left(\frac{2}{L}\right)}{\ln\left(\frac{z}{z_0}\right) - \psi_h\left(\frac{z}{L}\right)}, \tag{6}$$

where $T_g$ is the skin temperature, $T_a$ is the temperature from the lowest model level, $z_0$ is the roughness length, $L$ is the Obukhov length, and $\psi_h$ is the exchange coefficient for heat provided by the model's surface layer scheme. Therefore, the 2 m temperature is considering the state of the near-surface boundary layer and gives therefore more detailed information than, e.g., air temperature from the lowest model level. Still, we have to keep in mind that the expression calculating 2 m temperature in the model's surface layer scheme (Jiménez et al., 2012) relies on Monin-Obukhov similarity theory (MOST, Monin and Obukhov, 1954), which breaks down in katabatic flows (Grisogono et al., 2007) and is in need of substantial revision (Stiperski and Calaf, 2023). However, we consider its application reasonable in our case as we do not observe katabatic glacier winds in our time period of interest (Mott et al., 2020; Goger et al., 2022).

## 3 Upstream flow structure and gravity wave features in the glacier valley

The reference simulation (REF) is a real-case simulation of the glacier boundary layer from August 17, 2018. Under North-Westerly synoptic influence, a gravity wave formed over the North-Western ridge close to HEF, leading to a continuous disturbance of the glacier boundary layer. The case study day was dominated by cross-glacier flow and high values of non-stationarity (Mahrt, 1998) of the sensible heat flux during gravity-wave breaking episodes and a strong mesoscale influence on the glacier boundary layer. Due to the aforementioned phenomena, the local glacier boundary layer is heavily disturbed both in simulations and observations, and no katabatic down-glacier flow is present.

### 3.1 Spatial patterns of the wind field

A comparison of 10 m horizontal wind speeds between the REF, NO_UP and NO_GL simulations reveals differences in the flow structure over the glacier dependent on time of the day (Fig 2). In the morning (06:00, Fig 2a-c), cross-glacier flow with wind speeds of around $6\,\mathrm{m\,s^{-1}}$ dominates in all three simulations. In the REF simulation, this cross-glacier flow is present throughout the simulation. In the NO_UP and NO_GL simulations, however, we note the weakening of the cross-glacier flow, visible in reduced wind speeds after 08:00 (Fig 2e,f). While the cross-glacier flow prevails with reduced wind speeds in the REF simulation at 10:00 (Fig 2g), the NO_UP and NO_GL simulations show an up-glacier flow dominating the wind field at the glacier tongue (Fig 2h,i). This up-glacier flow persists until 12:00 (Fig 2k,l), together with a horizontal wind speed maximum at the North-facing slope next to the glacier.

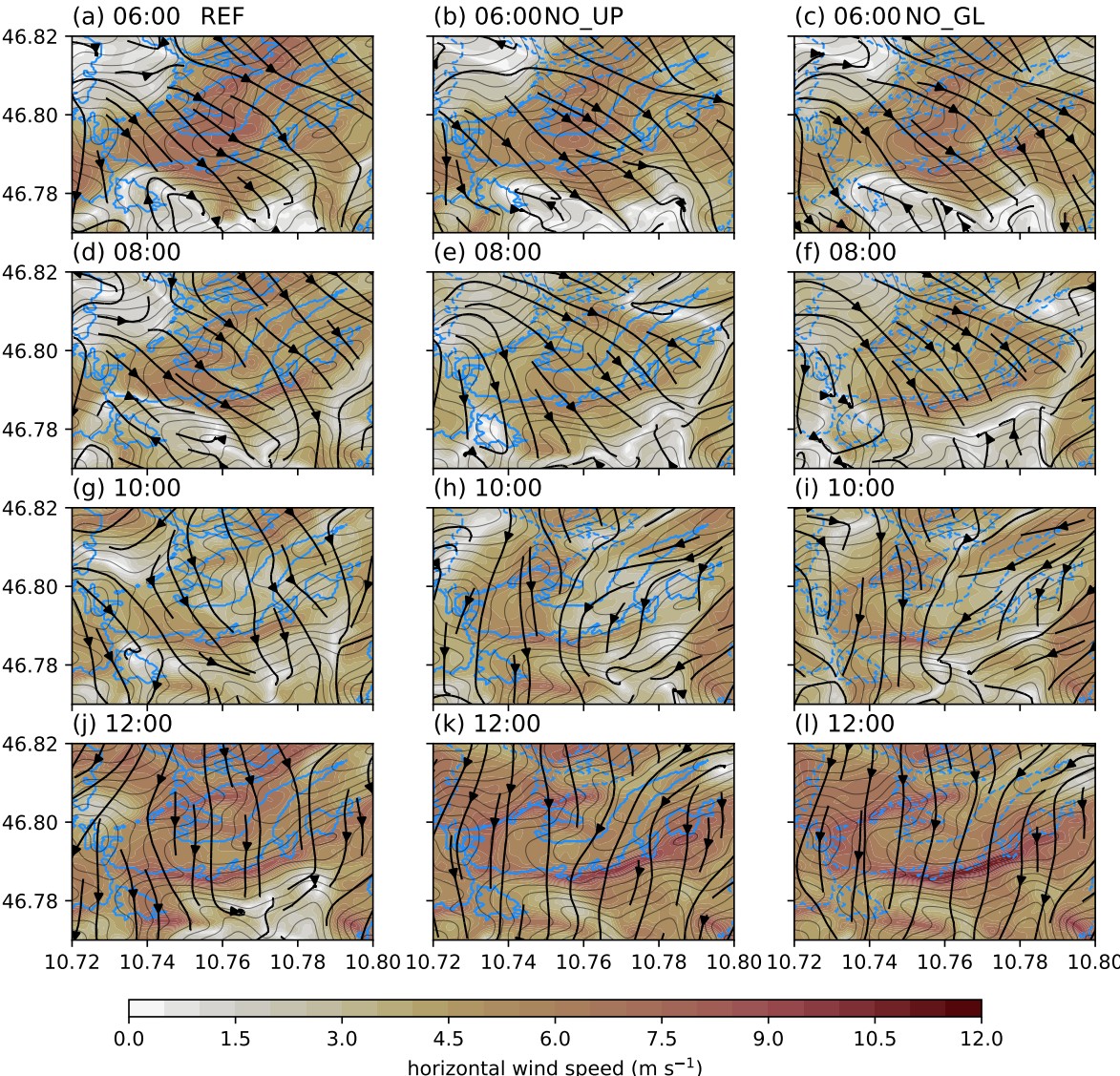

**Figure 2.** Simulated flow structure with 10 m horizontal wind speed (colors) and streamlines (black arrows) at four different times (06:00, a-c; 08:00, d-f; 10:00, g-i; 12:00, j-l). Left row: REF; middle: NO_UP, right row: NO_GL. The blue contours represent the glacier outlines in the simulations, while dashed blue lines indicate the location of the 'missing' ice surfaces. The thin black contours show model topography.

## 3.2 Vertical structure of the upstream flow and above HEF

Although the surface flow showed similar characteristics in the different simulations at 06:00, the vertical cross-sections show some marked differences. In the REF simulation, the cross-section of steepening isentropes at around 3500 m above mean sea level in the HEF valley at 06:00 reveals a gravity wave with hydraulic jump-like features (Fig. 3a), with an elevated turbulence

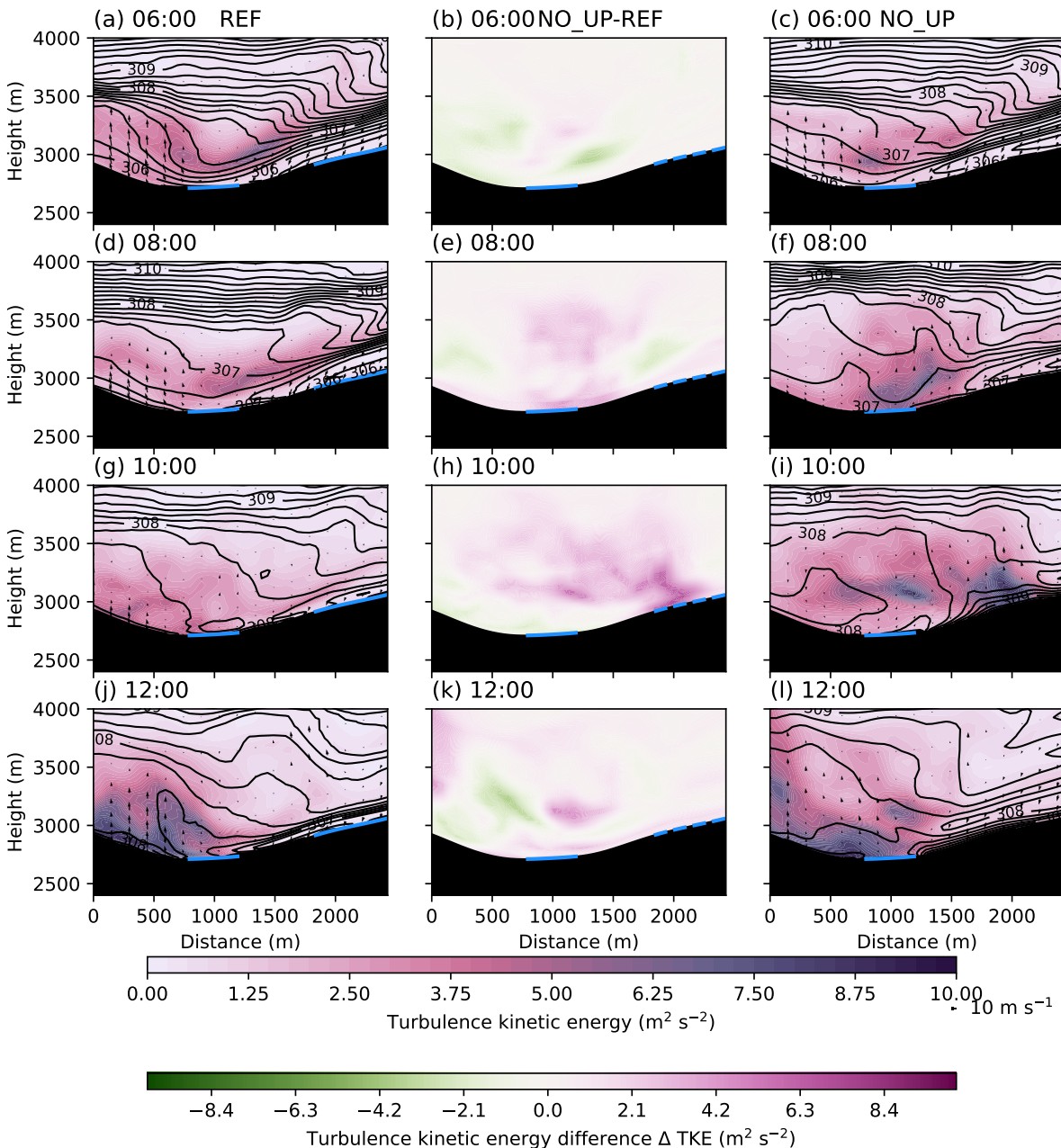

**Figure 3.** Vertical cross-section along the black line in Fig. 1b ("looking up" the valley) of simulated total TKE (subgrid + resolved, colors), isentropes (black contours), and cross-valley wind speed (arrows) from REF (left column), NO_UP (right column), and the difference in TKE between NO_UP-REF (middle column) at four different times (06:00, a-c; 08:00, d-f; 10:00, g-i; 12:00, j-l). The blue full lines show glacier surfaces, while dashed blue lines indicate the location of the removed glacier surfaces.

kinetic energy (TKE) maximum around 200 m above the surface. The overturning isentropes in the NO_UP simulation (Fig. 3c) indicate wave breaking with reduced near-surface static stability and a weaker TKE maximum ($\Delta$TKE=-5 m$^2$ s$^{-2}$), while the general structure of the gravity wave is similar to REF. Two hours later (08:00), the isentropes overturn in the REF simulation as well (Fig. 3d), while the gravity wave in NO_UP simulation already broke with reduced upstream stability under the influence of surface friction (Fig. 3f). The strong turbulent mixing over HEF tongue with higher TKE values is evident in NO_UP compared to REF (Fig. 3e). At 10:00, the gravity wave in REF is breaking as well (Fig. 3g), while NO_UP shows no distinct gravity wave pattern anymore according to the isentropes (Fig. 3i) with strong turbulent mixing and higher TKE values are present over HEF tongue (Fig. 3h). At 12:00, the highest TKE values are present over the glacier in REF, and the gravity wave re-established with a distinct dynamically forced downslope flow (Fig. 3j), while in the NO_UP simulation, reduced stability is visible upstream in association with a much weaker gravity wave (Fig. 3i) and reduced TKE values (Fig. 3k). To summarize, a gravity wave is present in both simulations, breaking over the HEF glacier valley. However, in the NO_UP simulation, the gravity wave is weaker and has a different breaking pattern, leading to higher TKE values and enhanced mixing over the glacier tongue. In the NO_GL simulations, the general structure and gravity wave formation is similar to that of the NO_UP simulation (Fig. A1). Therefore, we conclude that most of the gravity wave's dynamics is governed by the upstream glaciers and not by HEF itself.

Next we consider the time evolution of the vertical profiles of the atmosphere at the upstream location and at the HEF tongue. In all simulations at 06:00, the potential temperature profile at HEF tongue reveals a stable boundary layer (Fig. 4a) which differs from the stratification further aloft at both locations (HEF tongue and upstream), in accordance with a jet-like cross-glacier flow (Fig. 4e). The only difference between REF and the simulations without the upstream glaciers is the strength of the jet maximum. At the upstream location, North-Westerly flow is already established at 06:00 and potential temperature profiles are stably stratified in all simulations. The Scorer parameter (Eq. 1, Fig. 4i) shows, as expected, local maxima close to the inversions (cf. Fig. 4a) except for one peak in the REF simulation at the upstream location, favoring gravity wave formation.

Two hours later, differences start to emerge between the simulations: At the upstream location, the near-surface potential temperature difference is up to 4 K, and the profiles in the simulations without the upstream glaciers now show unstable stratification (and a convective boundary layer) compared to the neutral profile at this time in the REF simulation forming under the influence of the gravity wave. The Scorer parameter shows almost no favorable conditions for gravity wave formation, in accordance with the weakened gravity wave visible in the cross-sections at 08:00 (cf. Fig. 3b,f). At HEF tongue, potential temperature values within $\approx 600$ m above the surface are higher by 2 K in NO_UP and NO_GL compared to REF (cf. Fig. 4b). The wind speeds show a less distinct jet in the simulations without the upstream glaciers, while there is still a distinct low-level jet maximum in REF over HEF tongue (cf. Fig. 4f).

At 10:00, the flow is North-Westerly at upstream location in for all simulations, and the potential temperature profiles (cf. Fig. 4c) show a similar picture as at 08:00 at both locations, however, the wind direction over the glacier tongue now reveals a distinct up-valley flow in the NO_UP and NO_GL simulations (Fig. 4g), while the flow remains cross-glacier in the REF simulation. The Scorer parameter again shows favorable conditions for gravity wave formation, which are strongest in the REF simulation (Fig. 4k).

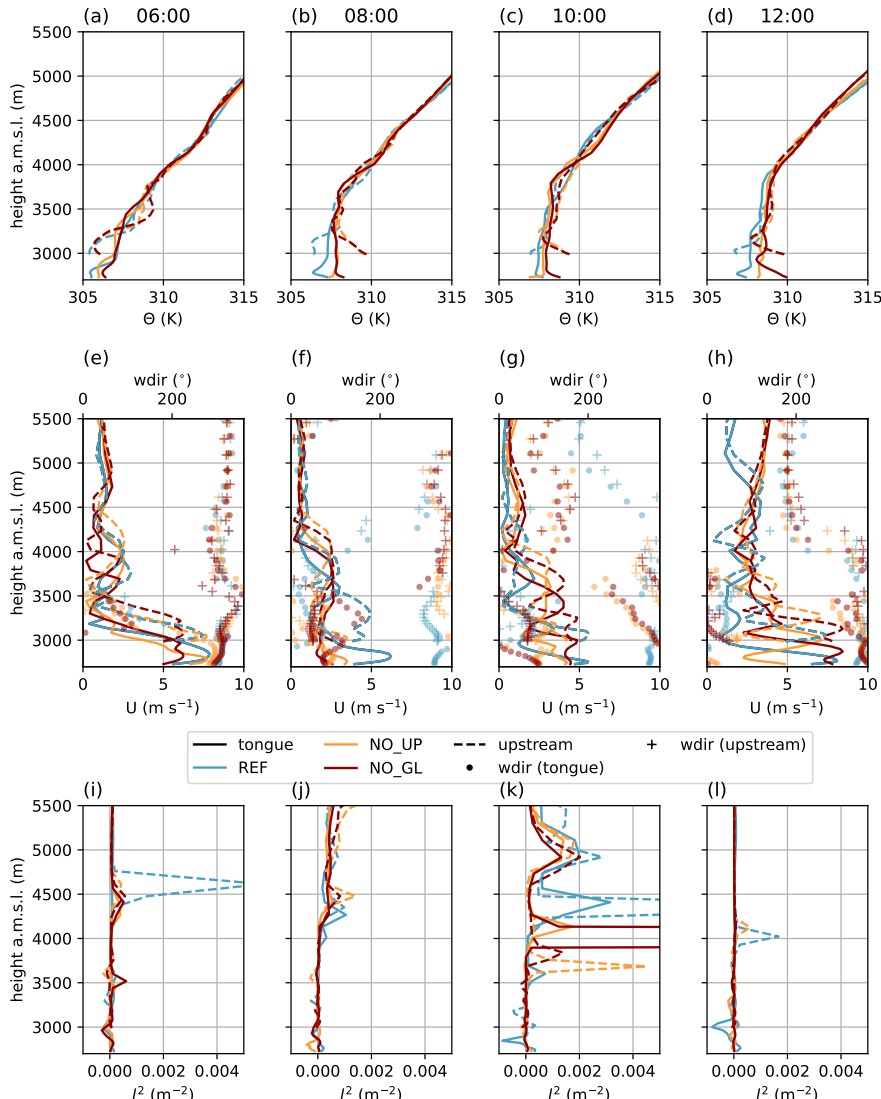

**Figure 4.** Vertical Profiles of potential temperature (panels a-d), horizontal wind speed and direction (panels e-h), and the Scorer parameter (panels i-l) from REF (blue), NO_UP (orange), and NO_GL (red) from two locations ("HEF tongue", full lines, red point in Fig. 1; and "Upstream", dashed lines, red star in Fig. 1) from four times.

Finally, at 12:00, there are clear differences in the potential temperature profiles close to the surface: At HEF tongue, we note a convective boundary layer in NO_GL, while there is mostly neutral stratification in REF and NO_UP (Fig. 4d). The wind patterns (Fig. 4h) show a chaotic behavior at HEF tongue, due to the gravity wave breaking and strong turbulence (cf. Fig. 3), while it is clear that the synoptic flow from aloft changes direction below 4000 m a.m.s.l., and all simulations now show an up-glacier flow. Upstream, there is still stable stratification in REF, while the simulations without the upstream glaciers

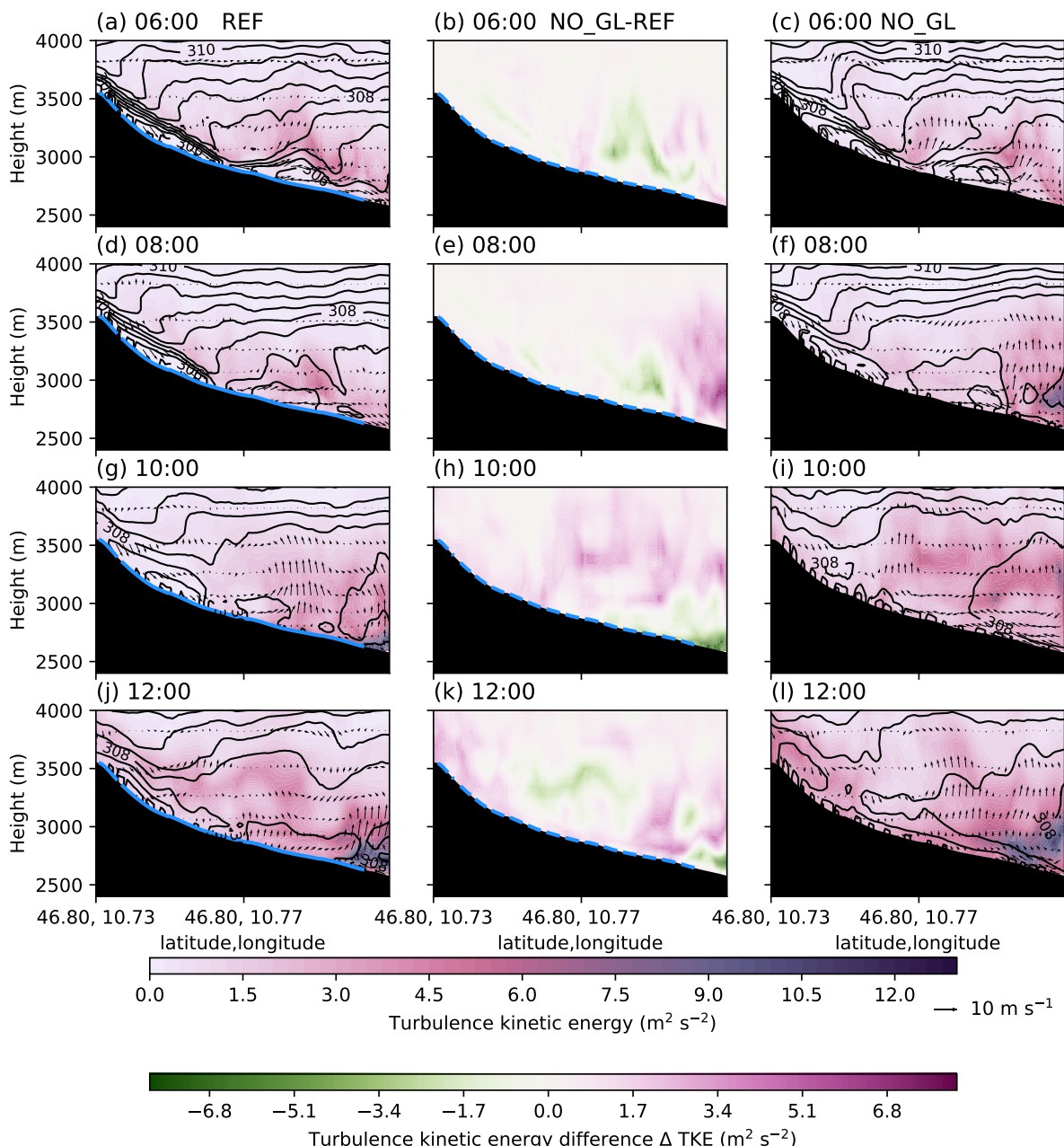

**Figure 5.** Vertical cross-section along the line in Fig. 1c of simulated total TKE (colors), isentropes (black contours), and along-valley wind speed (arrows) from from REF (left column), NO_GL (right column), and the difference in TKE between NO_GL-REF (middle column) at four different times (06:00, a-c; 08:00, d-f; 10:00, g-i; 12:00, j-l). The blue full lines show glacier surfaces, while dashed blue lines indicate the location of the removed glacier surfaces.

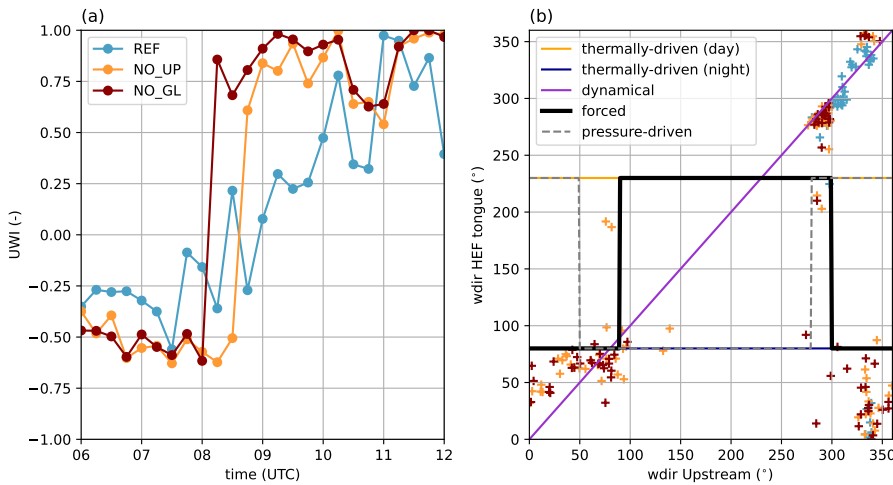

**Figure 6.** Panel a) Up-valley wind index (UWI) for REF, NO_UP, and NO_GL for the location HEF tongue. Panel b) Scatter plot of wind direction (+) at locations 'Upstream' (red star in Fig. 1) and at 'HEF tongue' (red point in Fig. 1) for all three simulations for our time period of interest. The lines in purple, black, and grey show the idealized categorizations from Whiteman and Doran (1993).

show neutral or convective boundary layers. The Scorer parameter again reveals no signs favorable for gravity wave formation (Fig. 4l).

Since we noted a distinct up-valley flow in the simulations without the upstream glaciers (NO_UP and NO_GL), we also explore the vertical structure of the atmosphere along the glacier (Fig. 5). In the REF simulation there is a strongly stable
boundary layer (SBL) at the upper parts of the glacier at 06:00 (Fig. 5a), with a down-glacier flow, while below 3000 m a.m.s.l., the weaker flow, reduced stability and higher TKE values, are associated with the strong cross-glacier flow over the tongue (Fig. 3). In NO_GL, the isentropes show that there is generally weaker stratification (Fig. 5c) in accordance with lower TKE values associated with the weaker gravity wave (Fig. 5b). Two hours later, REF still shows a with jet-like flow over the upper part of HEF (Fig. 5d), however, in NO_GL, the stable stratification over the missing glacier is continuously weakened
(Fig. 5f). Furthermore, NO_GL exhibits higher TKE values than REF below 3000 m a.m.s.l. (Fig. 5e), related to the earlier breaking cross-glacier gravity wave. The atmosphere above HEF is well-mixed in both REF and NO_GL at 10:00 and 12:00 (Fig. 5g-l), and the SBL is mostly eroded in both simulations, while the vertical profiles in the NO_GL simulation even suggest a convective boundary layer at the glacier tongue (Fig. 4d). This is likely one of the major reasons why the gravity wave is able to plunge and break into the glacier valley earlier in the NO_GL simulation than in REF, because there is a weaker (or no)
cold-air which has to be eroded by the upper-level flow (Haid et al., 2022). Furthermore, the up-glacier flow is also noticeable in NO_GL with enhanced turbulent mixing (Fig. 5i,l).

## 3.3 Wind direction shift in the glacier valley

In our simulations, the flow is cross-glacier for all three cases (Fig. 6a) before 08:00, related to the gravity wave present over the glacier tongue. However, after 08:00, the situation changes: In REF, there is a gradual shift in $UWI$ (Eq. 3) towards 1 until 12:00, suggesting a gradual weakening of the cross-glacier flow associated with the gravity wave. However, in the other two simulations, $UWI$ shows a sudden shift towards 1 at 08:00 (NO_GL) and 08:30 (NO_UP), suggesting up-glacier flow. When the gravity wave breaks (cf. Fig. 3, at 10:00), the up-valley flow is established (cf. Fig. 2) and, in contrast to the REF simulation, the gravity wave is unable to re-develop and the up-glacier flow persists after 09:00 in NO_UP and NO_GL.

According to Whiteman and Doran (1993)'s classification, the wind direction points in REF simulation collapse onto a diagonal line (Fig. 6b), suggesting that the wind structure in the HEF valley is dominated by downward momentum transport by the gravity wave. For the NO_GL and NO_UP simulations, a different picture emerges: Some of the points still follow the diagonal line (hence, downward momentum transport), but others are grouped in the lower left or lower right corner. This pattern corresponds to forced channelling, where the synoptic flow is channelled into the valley. In theory, pressure-driven channeling could also be a reason for the up-glacier flow (with a slightly different wind direction pattern), but this would require a horizontal pressure gradient along the valley axis. We calculated the horizontal pressure gradient (not shown), but it showed no distinct signal, hence we conclude that forced channelling is the major mechanism at play in the simulations without the upstream glaciers. Previous research on channelling flows in valleys suggests that shallow valleys are more prone to channelling of synoptic flows (Whiteman and Doran, 1993; Steyn et al., 2013) than deep Alpine valleys, where thermally-induced flows are very resilient to synoptic influence (Zängl, 2009). Given that the upper Rofen Valley, where HEF is located, is rather shallow compared to its surroundings (the height difference between HEF tongue and the upstream location is around 500 m), channelling of synoptic cross-glacier flows is a realistic scenario. To summarize, we conclude that the gravity wave leads to downward momentum transport over the HEF valley, and when it breaks and henceforth weakens, the synoptic flow is channelled via forced channelling into the glacier valley. In this case study, the up-glacier flows after 09:00 can be explained with dynamical forcing and the thermal forcing component (e.g., slope or valley flows due to differential heating) is negligible.

## 4 Impact on glacier boundary layer

In this Section, we compare the three simulations with respect to sensible heat fluxes, advection patterns, the heat budget, and 2 m temperature. While we cannot resolve the glacier boundary layer in great detail, we are interested in assessing the impacts of the changed ice surfaces on the valley atmosphere and surface processes above HEF. Since we conduct semi-idealized simulations, we assess the changes in the sensitivity runs (NO_UP and NO_GL) compared to the reference simulation, although we are aware that the sensible heat flux is simulated too strongly in the REF simulation compared to the observations, but we also note that the general physical processes are simulated correctly (Goger et al., 2022).

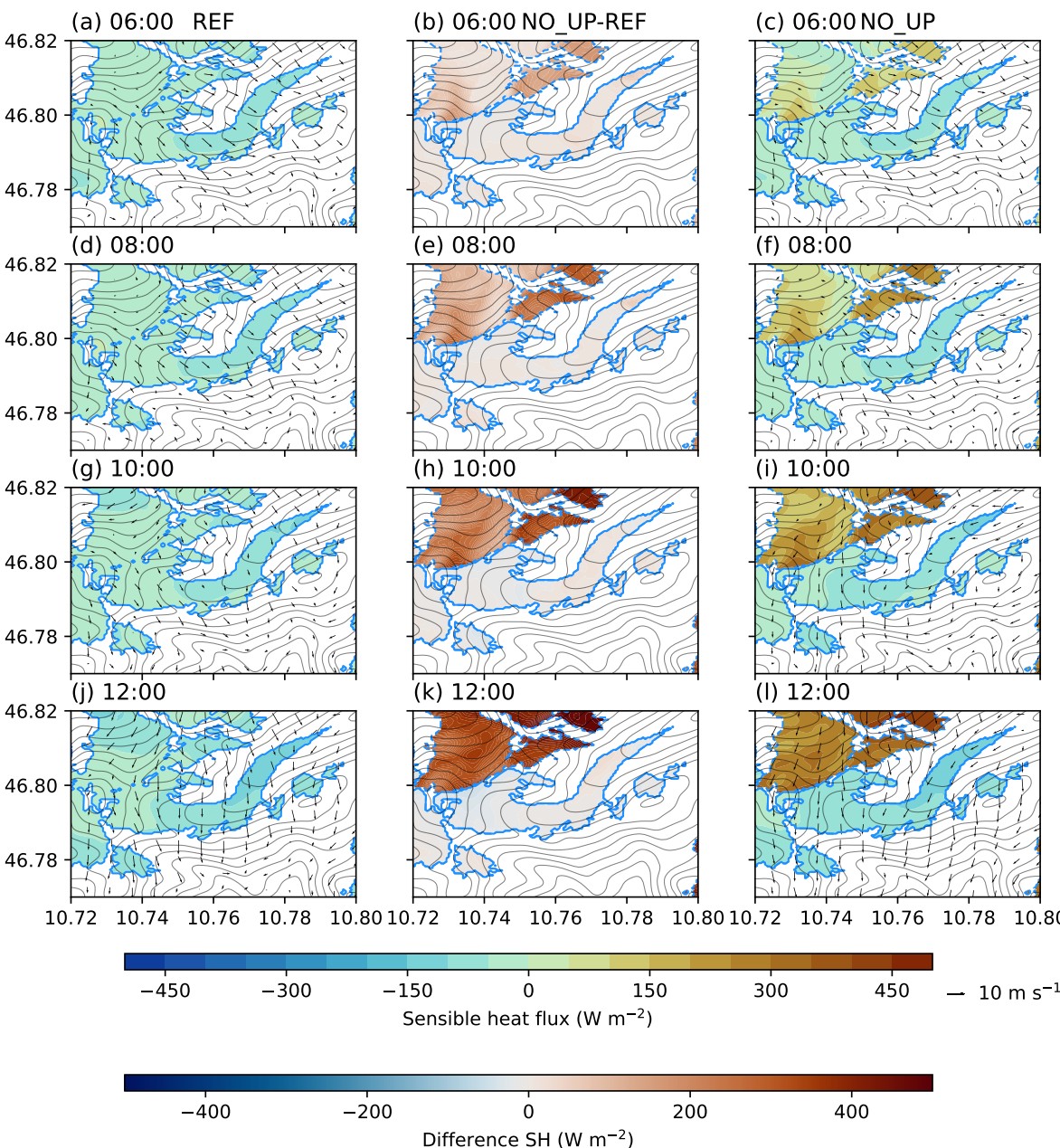

**Figure 7.** Simulated sensible heat flux (colors) from the lowest model level and 10 m wind speed (black arrows) at four different times (06:00, a-c; 08:00, d-e; 10:00, g-i; 12:00, j-l). Left row: REF; middle: Difference between REF and NO_UP, right row: NO_UP. The blue contours represent the glacier outlines in the simulations, while dashed blue lines in the middle row indicate the 'missing' ice surfaces. The thin black contours show model topography.

## 4.1 Spatial patterns of sensible heat flux

The surface sensible heat flux plays a pivotal role in the energy exchange over glaciers in the summer months. Over a melting glacier the surface temperature is constant at 0°C, impacting the bulk formulation of the sensible heat flux (Stull, 1988, their equation 7.4.1d). As the surface temperature over a melting glacier is exactly 0°, sensible heat fluxes strongly depend on the wind speed, we can expect that changing wind patterns drive changes in the sensible heat flux structure. In the REF simulation with the realistic glacier surfaces, the sensible heat flux is negative (from the atmosphere to the ice, atmospheric notation) during the entire simulation over HEF (Fig. 7a,d,g,j). Over the remaining ice surface (i.e., HEF) in NO_UP, sensible heat flux magnitudes are by $50\,\mathrm{W\,m^{-2}}$ smaller compared to REF at 06:00 (Fig. 7b). The changed sensible heat fluxes are present during the entire simulation time at the remaining ice surface in NO_UP. Interestingly, the sensible heat flux difference between NO_UP and REF is very small when cross-glacier flow is present in both simulations at the upper part of HEF (Fig. 7h,k,i,l). The largest differences between REF and NO_UP are visible at 12:00, when the gravity wave broke and the strong up-glacier flow is present in NO_UP (Fig. 7i,l). It is not surprising that the sensible heat fluxes over the missing ice surfaces change sign between REF and NO_UP (Fig. 7k). This indicates that the removing upstream ice surfaces, and therefore their influence on atmospheric flow structures impacts spatial variability in sensible heat fluxes and consequently can be expected to have an impact on HEF's melting patterns.

When we remove all glaciers from the model domain (Fig. 8), sensible heat fluxes reveal a large difference between NO_GL and REF changing over simulation time. At 06:00, sensible heat fluxes exhibit a difference of around $100\,\mathrm{W\,m^{-2}}$ over the missing ice surfaces (Fig. 8b), while with progressing simulation time these differences change up to more than $500\,\mathrm{W\,m^{-2}}$ with opposite signs (Fig. 8k). This is not surprising, since changing the land-use category in the model from "snow or ice" to "bare rock" leads to changes in albedo and roughness length, (Tab. 1), henceforth leading to positive sensible heat fluxes during the daytime. Interestingly, sensible heat fluxes remain smaller in the glacier valley than its surroundings in the NO_GL simulation (Fig. 8c,f,i,l), likely due to the sheltered location of the valley and topographic shading. However, despite the large differences in sensible heat flux over missing HEF, removing all ice surfaces does not have a large impact on the general flow structure, given the simulated 10 m wind speeds (Fig. 8) and the very similar gravity wave structure as in NO_UP. Therefore, we can finally conclude that HEF's ABL structure is in this particular case study mostly dominated by the upstream formation of gravity waves, as described in Section 3, and HEF itself plays a secondary role. Still, observations suggest the persistence of a very shallow katabatic flow even in highly disturbed conditions (Mott et al., 2020), which is unable to be resolved by the model.

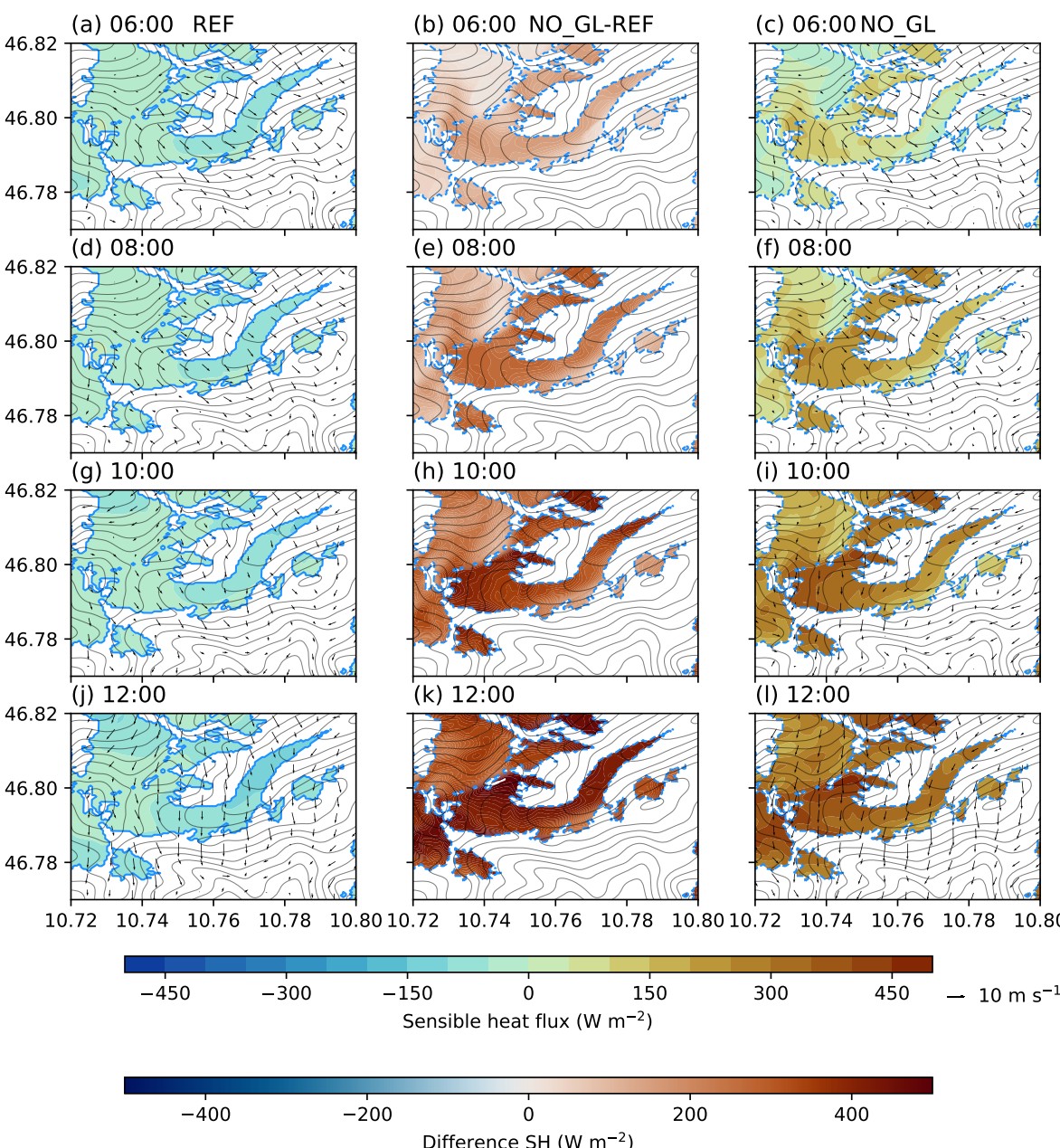

**Figure 8.** Simulated Sensible heat flux (colors) from the lowest model level and 10 m wind speed (black arrows) at four different times (06:00, a-c; 08:00, d-e; 10:00, g-i; 12:00, j-i). Left row: REF; middle: Difference between REF and NO_GL, right row: NO_GL. The blue contours represent the glacier outlines in the simulations, while dashed blue lines indicate the location of the 'missing' ice surfaces. The thin black contours show model topography.

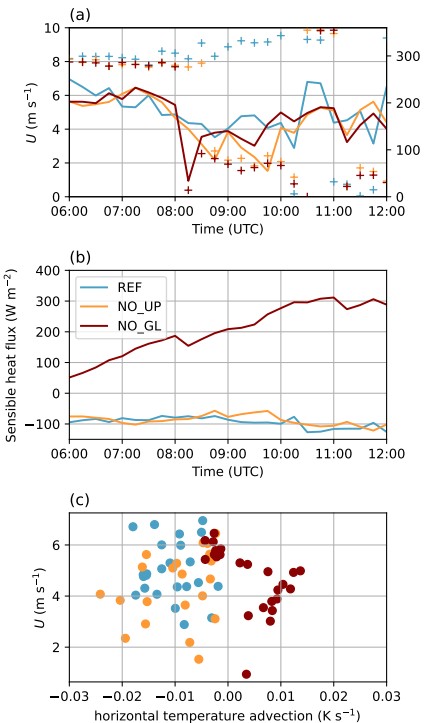

**Figure 9.** Panel a) Time series of horizontal wind speed (lines) and wind direction (+) from the lowest model level for REF (blue), NO_UP (orange), and NO_GL (red) simulations at HEF tongue. Panel b) as panel a), but for sensible heat flux. Panel c) Scatter plot of horizontal temperature advection and horizontal wind speed over the time period of interest (06:00-12:00).

## 4.2 Horizontal advection patterns and heat budget

Time series of horizontal wind speed and direction from the lowest model level at HEF tongue reveal no substantial differences in wind speed magnitude between the three simulations (Fig. 9a) despite the differences in wind direction between REF and the sensitivity simulations, related to the up-valley flow caused by earlier gravity wave breaking, as outlined in the previous sections. With no change of wind speed, the sensible heat flux magnitude at HEF tongue (Fig. 7) does not change significantly (less than $50\,\mathrm{W\,m^{-2}}$) between REF and NO_UP either (Fig. 9b), however, when HEF is removed in NO_GL sensible heat flux changes sign, as expected, and increases substantially in magnitude, highlighting the importance of the glacier surface for the local sensible heat exchange.

Regarding the horizontal advection patterns over HEF (Eq. 4), Goger et al. (2022) showed that they are strongly influenced by the wind speed and dominating wind direction. As in the REF simulation, the glacier tongue in the NO_UP simulation is under the influence of horizontal cold-air advection during our time of interest (Fig. 9c). In the REF simulation, the horizontal cold-air advection is mostly associated with the gravity wave advecting potentially colder air from the upstream glaciers towards HEF tongue. The horizontal advection patterns are very similar in the REF and NO_UP simulations, suggesting

that the katabatic mechanism is not important in the simulations (Fig. 9c). In the NO_GL simulation, on the other hand, the advection is generally warm, despite the flow being cross-glacier. In general, gravity waves lead to warm-air advection due to

350 the isentropic drawdown, and because upstream glaciers are missing, this effect is expected over HEF tongue.

The heat budget (Eq. 5) of the REF simulation (Fig. 10a) reveals cooling of the column above HEF tongue between 06:00 and 08:30 with slight interruptions above 2800 m a.m.s.l. and oscillations in time. This coincides with the gravity wave breaking pattern, because the brief periods of positive $\frac{\partial \theta}{\partial t}$ correspond to the gravity wave breaking in REF (cf. Fig. 3), and the atmosphere above HEF experiences cooling from 09:00 until 12:00 at heights above 2850 m a.m.s.l. The time-averaged vertical heat budget

reveals that cold-air advection dominates during our time period of interest, as the gravity wave advects cold air from the upstream glaciers towards HEF tongue (Fig. 10d). In the NO_UP simulation, a different picture emerges (Fig. 10b): HEF tongue is well-mixed, with very weak cooling or heating, and with weaker absolute heating/cooling values than REF. This alternating pattern in NO_UP's heat budget leads to an overall zero net effect (Fig. 10e). Similar patterns are visible in the NO_GL simulation (Fig. 10c), but with stronger heating effects than the other simulations (Fig. 10f). We relate this to the

much weaker gravity wave also breaking earlier in the simulations without the upstream glacier surfaces (NO_UP and NO_GL) having a large influence on the heating rate over HEF tongue. To summarize, the vertical heating rate in REF shows distinct heating or warming patterns in relation to the gravity wave, where in NO_UP, there is a net zero effect, and in NO_GL shows a small warming effect.

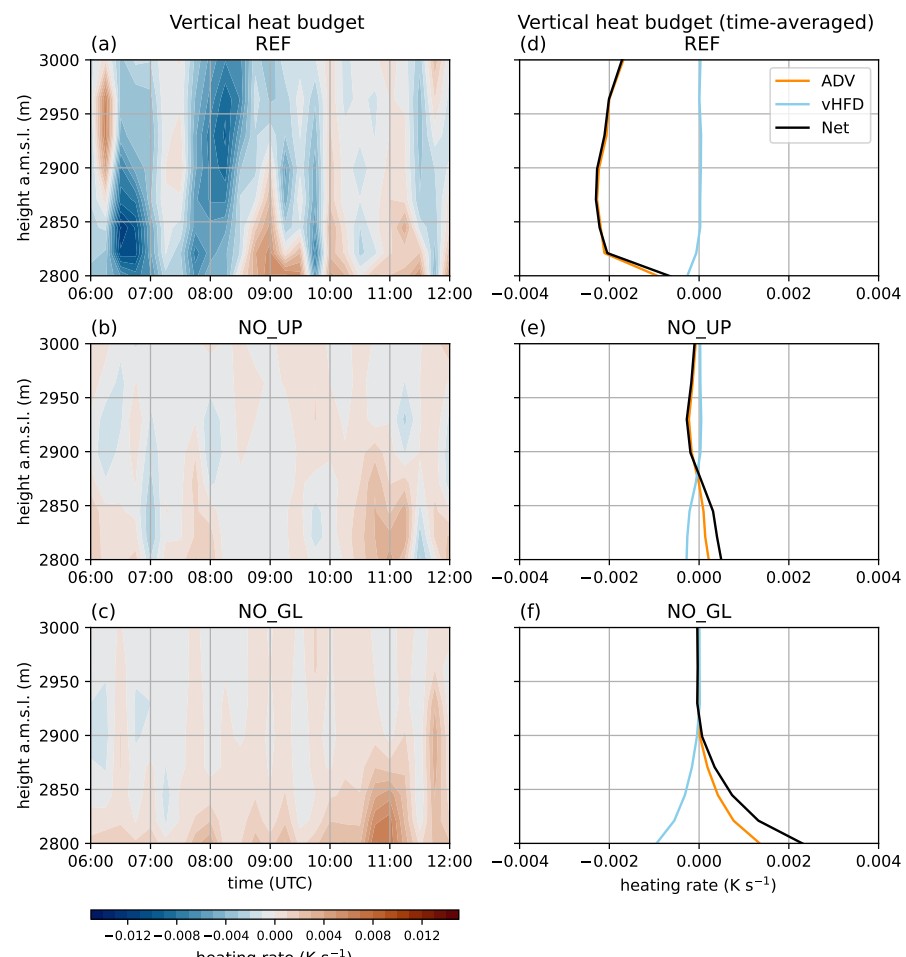

**Figure 10.** Panels a)-c) Time series of the total heat budget over HEF tongue for REF (a), NO_UP (b), and NO_GL (c). Panels d)-f) Time-averaged (over the time period from 06:00-12:00) components of the total vertical heat budget at HEF tongue (Net), namely temperature advection (ADV) and vertical heat flux divergence (vHFD), for REF (d), NO_UP (e), and NO_GL (f).

## 4.3 Resulting temperature structure on the glacier

The 2 m temperature (Eq. 6), averaged over our time period of interest (06:00–12:00) shows temperatures close to 0°C over all ice surfaces, especially over the upstream glaciers (Fig. 11a), while the glacier surroundings are up to 10°C warmer. The NO_UP simulation shows higher 2 m temperatures over the missing upstream glaciers with a temperature contrast of up to 5°C compared to REF (Fig. 11b). Furthermore, HEF tongue is warmer by 1°C than the REF counterpart (Fig. 11d), suggesting that the glacier tongue is influenced by the (warmer) surroundings (cf. Fig. 9). Finally, removing all glaciers (Fig. 11c) leads

to a temperature increase of 5°C at all missing ice surfaces compared to REF. The surroundings are 1°C warmer than in REF (Fig. 11f), suggesting a large influence by the strong turbulent mixing induced by the breaking gravity wave.

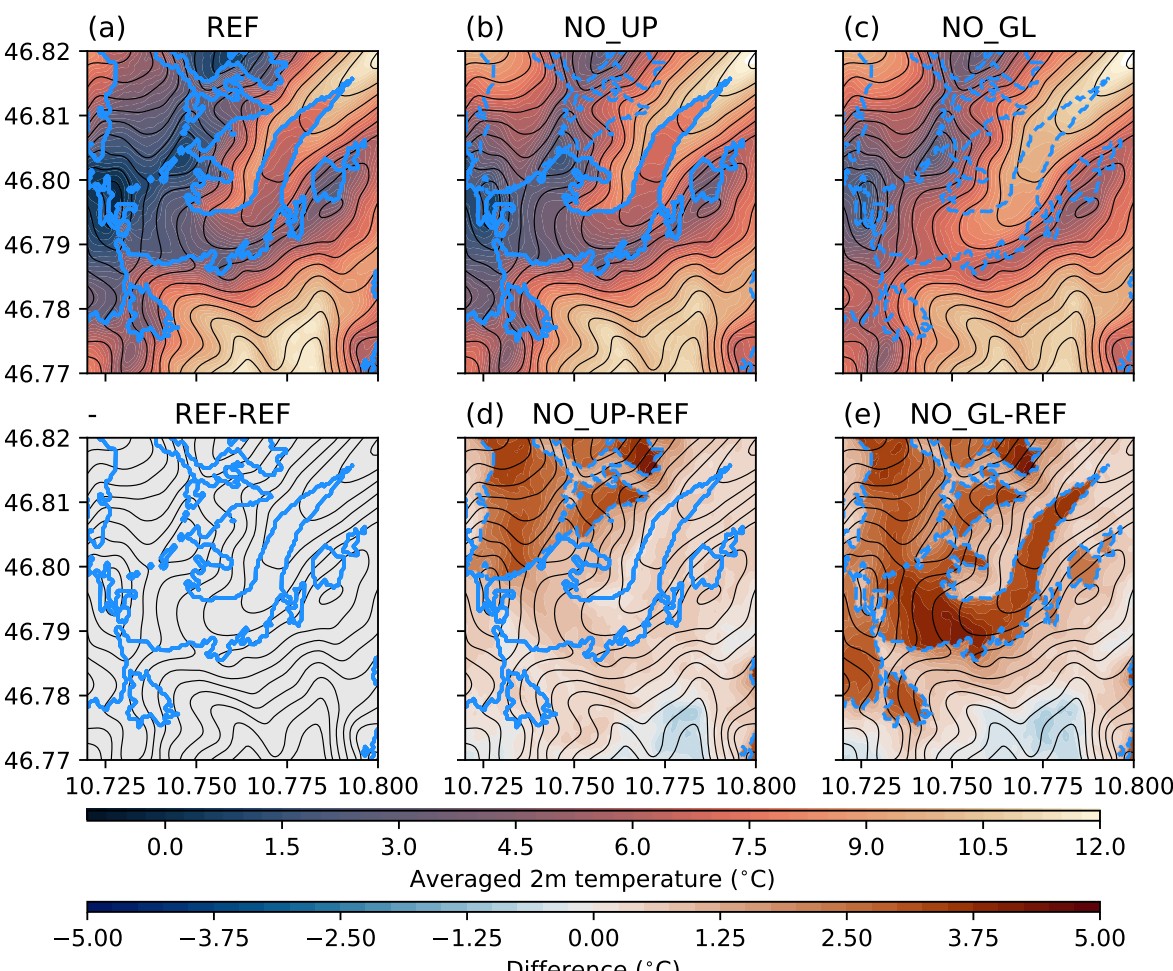

**Figure 11.** Panels a-c) Averaged 2 m temperature (colors) over the simulation period of interest (06:00–12) for REF, NO_UP, and NO_GL, respectively. Panels d-e) Difference in averaged 2 m temperature from REF. All panels: Topography (black contours), ice surfaces (blue outlines), missing ice surfaces compared to REF (red outlines).

## 5   Discussion

In general, our LES simulations proved an ideal test bed to study the impact of dynamically-induced flows on the glacier boundary layer. As already discussed in Goger et al. (2022) and Voordendag et al. (2024), more than 10 grid points across the glacier valley are present in our domain, and this hints that the major MoBL processes are resolved (Wagner et al., 2014). When we assume that effective horizontal resolution of the model is $7\Delta x$ (Stull, 2017, their pg. 761), phenomena of a horizontal extent of 336 m are resolved on the grid, which is adequate to resolve the gravity wave across the valley and the associated processes. Since the focus of the study is on the dynamical forcing and gravity waves, which form on scales larger than the

small-scale glacier boundary layer, we can expect that the model delivers reliable results on these processes. Still, the very near-surface flow is influenced strongly by the subgrid-scale parametrization, and the potential very shallow katabatic flow that might persist very close to the surface (cf., Mott et al., 2020) is not captured by the model.

We analyzed results from a six-hour period (06:00-12:00), not only because REF started to deviate from observations after 12:00 (Goger et al., 2022), but also, because after this time the two sensitivity simulations (NO_UP and NO_GL) did not give new insights on the gravity wave dynamics: In both NO_UP and NO_GL, the gravity wave does not re-establish itself and up-glacier flow prevails as in the REF simulation after 12:00 (not shown). While we only show findings from a single case study, it indicates that HEF cannot be treated in isolation from the surrounding glacier surfaces, because of the strong impact of the upstream glaciers on gravity wave formation. This is similar to findings from other studies in different glacierised settings. For example, a study of foehn flows over the Larsen C ice shelf, Antarctica (Turton et al., 2018) found that the stabilizing effect of upstream ice surfaces influences the isentrope drawdown of gravity waves. Jonassen et al. (2014) simulated gravity waves over the Hofsjökull icecap, Iceland, and also found stronger downslope flow acceleration in simulations with the icecap, while removing the icecap led to weaker downslope flows.

The gravity wave in both the NO_UP and NO_GL simulations is weaker and breaks earlier compared to the simulation with realistic glaciers (REF), leading to stronger turbulent mixing. The dynamical force balance due to gravity wave breaking changes the flow regime inside the valley from across glacier caused by downward momentum transport in REF, towards an up-valley flow due to forced channeling (Whiteman and Doran, 1993). Another interesting aspect is that the flow structure in the simulations without the upstream glaciers (NO_UP and NO_GL) is very similar, *despite* the presence of the HEF glacier tongue in NO_UP. This suggests that in our particular case study the flow structure in the glacier valley is so strongly dominated by the upstream conditions and that local effects of the HEF ice surface itself do not provide a relevant feedback to the simulated local wind patterns. On the other hand, the HEF ice surface has a dominant control on the surface heat exchange.

In the two sensitivity simulations, the earlier gravity wave breaking leads to stronger mixing above the HEF glacier surface and stronger sensible heat fluxes into the glacier. Therefore, we can conclude that removing upstream glaciers leads to an overall destabilization of the atmosphere above HEF, increase of near-surface air temperatures, and weaker gravity waves. The role of the upstream glaciers on the cross-glacier flow is non-negligible, therefore, it makes sense to speak of a *system* of glaciers (HEF and the upstream glaciers) influencing each other's local microclimates on a scale of around 5-10 km. However, this 'length scale' for the development of mesoscale ice breezes (similar as in Conway et al., 2021) depends on the size of the glaciers and connected upstream ice fields. Examples for larger icefields influencing the local wind patterns on glacier tongue would be the Columbia icefield in Canada (Conway et al., 2021), the Jostedalsbreen icefield in Western Norway (Haualand et al., 2024), or Vatnajökull icecap in Iceland (Björnsson et al., 2005). All of them span more than 50 km, and dependent on flow direction, icefield breezes or gravity waves are favorable to develop. Our work allowed us to shed light on the impact of upstream icefields/glaciers on local glacier boundary layers, but more detailed research in combination with a wind climatology and flow-resolving simulations are necessary in the future.

A final open question to discuss is how representative our 6 hours of simulation are for HEF and its surroundings. Currently, we can only compare to a wind climatology at HEF compiled by Obleitner (1994), and they found a significant Northerly

gradient wind influence on the South-facing slope of the valley, the same wind direction as in our case study. Furthermore,
Mott et al. (2020) noted in the HEFEX campaign that in 20% of their wind observations the katabatic flow was 'disturbed'
and the glacier boundary layer was eroded. Therefore, we can assume that the described situation of strong North-Westerly
winds and gravity waves eroding the glacier boundary layer is not a single occurrence. Still, an updated wind climatology over
HEF is necessary to quantify these events. Furthermore, applying an intermediate complexity model such as HICAR (Reynolds
et al., 2023, 2024) to the region for entire seasons would shed more light on the typical wind patterns over HEF given its small
computational cost. However, we still see high-resolution, full-physics numerical simulations (e.g., Sauter and Galos, 2016;
Gerber et al., 2018; Mott et al., 2019; Draeger et al., 2024; Voordendag et al., 2024; Haualand et al., 2024) as the current
standard to investigate the physical processes over glaciers located in complex terrain despite their high computational cost.

## 6    Conclusions

We conducted semi-idealized large-eddy simulations at $\Delta x =$48 m over the Hintereisferner (HEF) glacier in the Austrian Alps
for a short case study with North-Westerly synoptic flow. The reference simulation (REF) was run with realistic glacier surfaces,
while the NO_UP simulation ran without upstream glaciers, and in the NO_GL simulation all ice surfaces were removed from
the domain. The results allow us to draw the following conclusions:

- Under North-Westerly synoptic flow, a gravity wave forms over HEF's tongue location in all three simulations, which,
  when it breaks leads to turbulent mixing over the HEF glacier surface.

- Removing the upstream glaciers in the NO_UP simulation leads to a neutral stratification with an unstable surface layer
  and henceforth less favorable conditions for gravity wave formation. The gravity wave in the NO_UP simulation is
  weaker than in REF and breaks 30 minutes earlier, leading to earlier onset of strong turbulent mixing over the remaining
  glacier surface and changing the cross-glacier flow towards up-glacier flow. A similar pattern is present in the NO_GL
  simulation.

- The upstream glaciers are not *necessary* for gravity wave formation, but *strongly* influence their strength by stabilizing
  the upstream profile.

- In this case study, the wind patterns over the glacier valley are governed by downward momentum transport by the
  gravity wave. When it breaks, the synoptic flow is channeled into the valley leading to up-glacier flows.

- Due to the changed gravity wave structure, higher temperatures, a higher spatial variability in sensible heat fluxes are
  present at HEF tongue location in the NO_UP and NO_GL simulations.

- Due to the strong gravity wave, the heat budget over HEF is negative in the REF simulation. However, when the gravity
  wave is weakened in the sensitivity runs, we note a net-zero heat budget in NO_UP and slight overall warming in
  NO_GL.

- The local boundary layer flow structure in the NO_GL simulation is very similar to the NO_UP simulation, suggesting that the atmosphere over HEF is therefore less governed by the surface below (ice), but rather by dynamical forcing.

- The results suggest that HEF is not isolated from its nearby environment (rock surface/slopes), and that the influence of upstream ice surfaces on local boundary-layer development cannot be disregarded. Henceforth, in the near future it is advisable to investigate a *system* of glaciers instead of studying processes on isolated glacier tongues only.

The present study gave insight to the impact of ice surfaces on gravity wave formation and breaking affecting glacier boundary layer development. In the future, similar studies with different upstream conditions could be conducted. However, one of the open questions in glacier boundary layer dynamics are the future changes in glacier boundary-layer structure in a warming climate, which could be quantified with high-resolution LES using future ice surfaces derived from climate projections.

*Code and data availability.* The WRF v4.1 model code can be downloaded from github (WRF, 2019), and the averaging module `WRF LES diagnostics` is available at Umek (2020). The model output is available upon request from BG. Figures were generated with `python-matplotlib` (Hunter, 2007) using colormaps by Crameri (2023).

*Author contributions.* BG set up and conducted the numerical simulations and wrote the initial draft of the manuscript. MO performed the analysis of numerical data within his internship at Universität Innsbruck in summer 2023 supervised by IS. IS and LN provided input on the analysis and all authors read and improved the manuscript where necessary.

*Competing interests.* The authors declare no competing interests.

*Acknowledgements.* This work is part of the project "Measuring and modeling snow-cover dynamics at high resolution for improving distributed mass balance research on mountain glaciers", a joint project fully funded by the Austrian Science Foundation (FWF, project number I 3841-N32, https://dx.doi.org/10.55776/I3841) and the Deutsche Forschungsgemeinschaft (DFG; project number SA 2339/7-1). The computational results presented have been achieved using the Vienna Scientific Cluster (VSC) under project number 71434. The work of I. Stiperski was funded through the European Research Council (ERC) under the European Union's Horizon 2020 research and innovation program (Grant agreement No. 101001691). We thank Michael Haugeneder and Cole Lord-May for their concise and thoughtful referee reports leading to the improvement of the manuscript.

## Appendix A: Vertical cross-glacier (NO_GL simulation) and along-glacier (NO_UP simulation) cross-sections

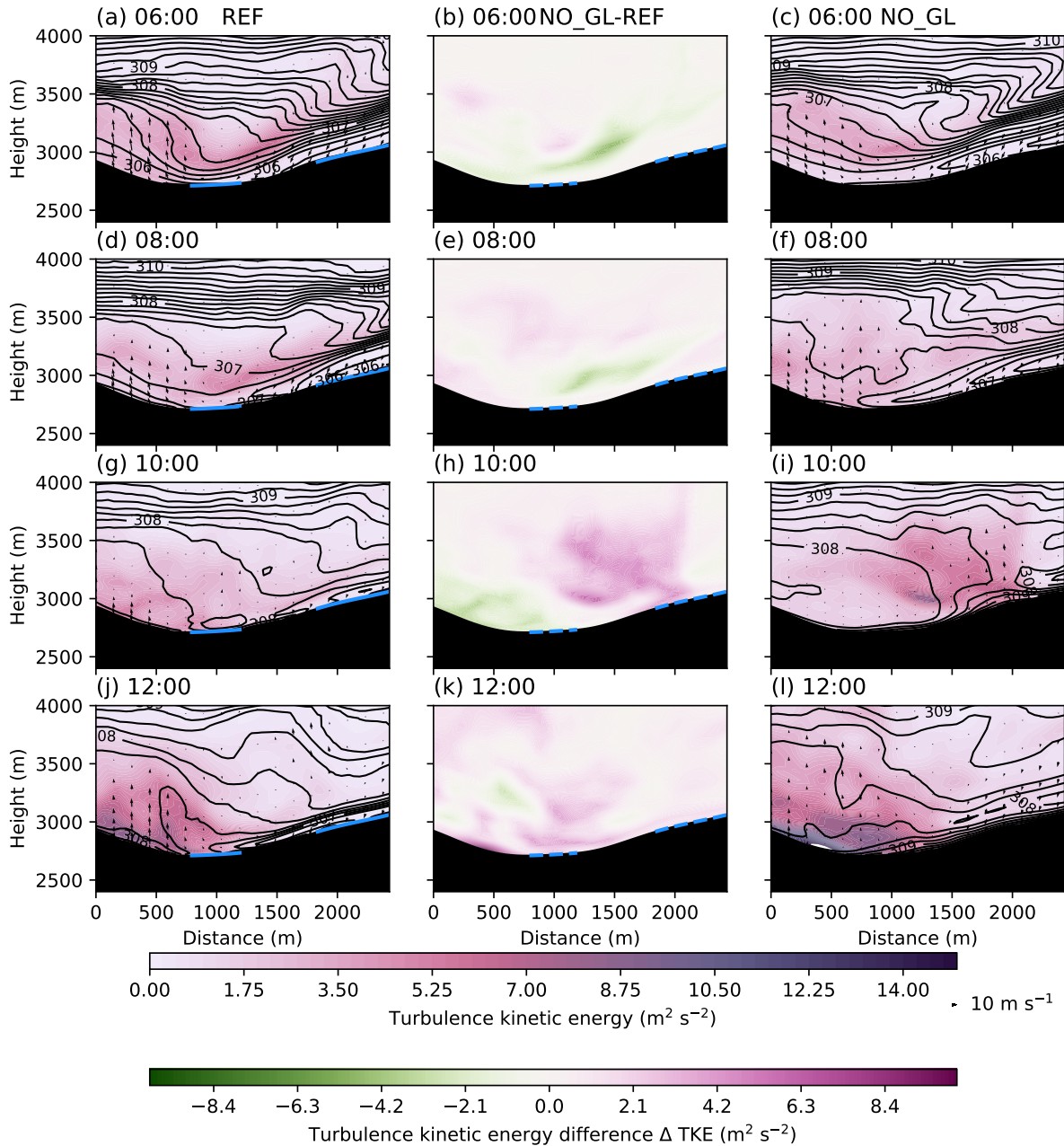

**Figure A1.** Vertical cross-section along the black line in Fig. 1b ("looking up" the valley) of simulated total TKE (subgrid + resolved, colors), isentropes (black contours), and cross-valley wind speed (arrows) from REF (left column), NO_GL (right column), and the difference in TKE between NO_GL-REF (middle column) at four different times (06:00, a-c; 08:00, d-f; 10:00, g-i; 12:00, j-l). The blue full lines show glacier surfaces, while dashed blue lines indicate the location of the removed glacier surfaces.

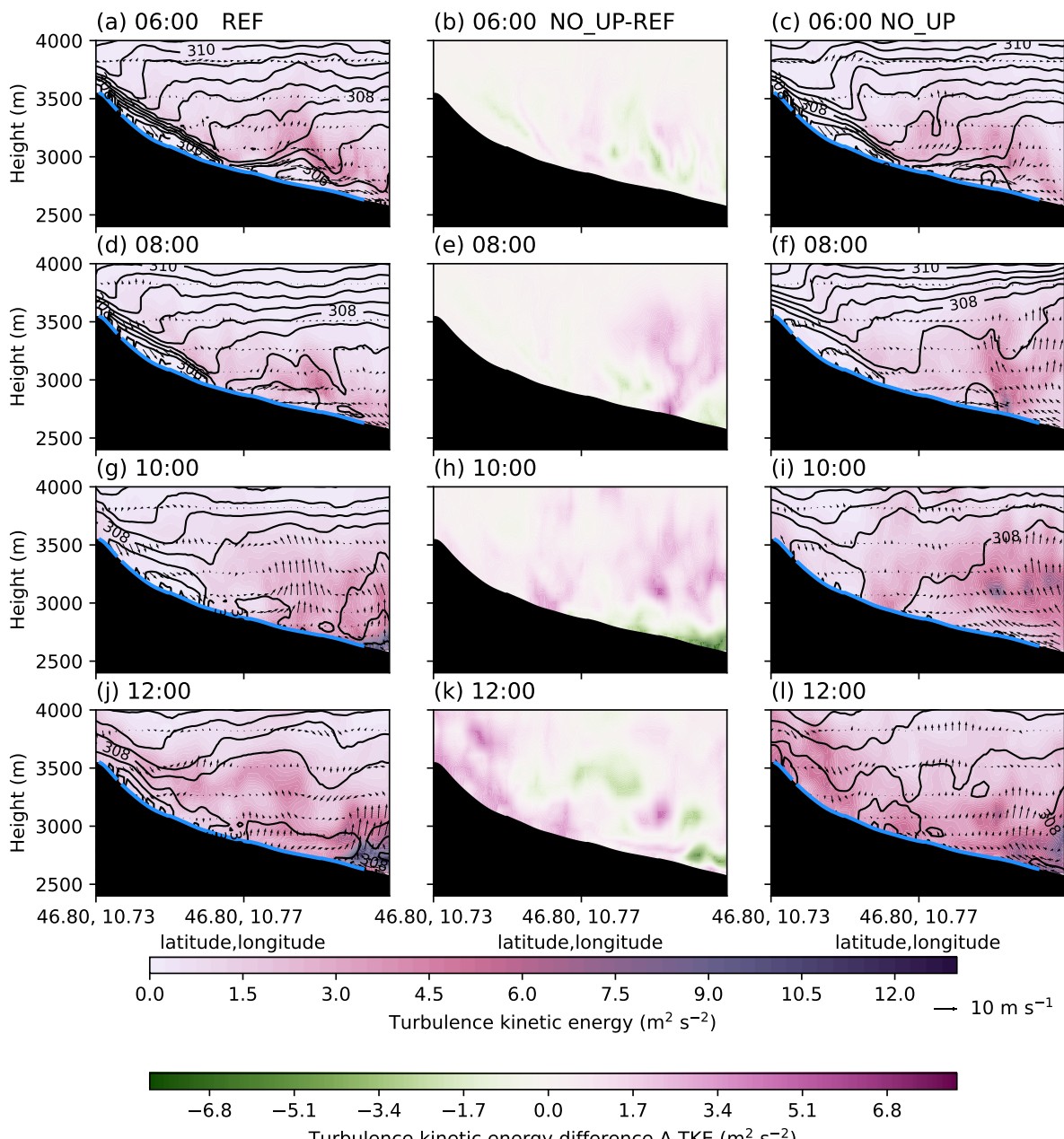

**Figure A2.** Vertical cross-section along the line in Fig. 1c of simulated total TKE (colors), isentropes (black contours), and along-valley wind speed (arrows) from from REF (left column), NO_UP (right column), and the difference in TKE between NO_UP-REF (middle column) at four different times (06:00, a-c; 08:00, d-f; 10:00, g-i; 12:00, j-l). The blue full lines show glacier surfaces, while dashed blue lines indicate the location of the removed glacier surfaces.

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
