# Peer review of "Investigating the influence of changing ice surfaces on gravity wave formation impacting glacier boundary-layer flow with large-eddy simulations"

_EGUsphere, 2024_

## Referee Comment (RC1)

**Review of "Investigating the influence of changing ice surfaces on gravity wave formation and glacier boundary-layer flow with large-eddy simulations" by Goger et al.**

Michael Haugeneder, michael.haugeneder@slf.ch

September 5, 2024

The authors present a case study of large-eddy simulations of the atmosphere above the highly heterogeneous high-mountain terrain surrounding Hintereisferner (HEF). They compare a reference simulation featuring realistic ice surfaces with two sensitivity analyses comprising different surface properties. In one run, they only replace the upstream glacier surfaces, while in the other sensitivity run, all glacier surfaces are replaced by bare ground. With these different setups, the authors aim at investigating the influence of upstream surface properties on the flow dynamics over the HEF ultimately affecting the heat fluxes in the near-ice atmosphere at HEF.

Their work contributes significantly to understanding the larger-scale (and non-local) influences on the local flow field. Comparing runs with modified surface properties to verified "real-world" runs offers promising insights advancing the current understanding of process interactions at a wide range of scales. However, I suggest revising the presentation of the results to facilitate better understanding.

If the authors have any questions, please don't hesitate to ask.

**Major Comments**

1. language and spelling: The manuscript will strongly profit from a thorough revision of the language including syntax and spelling. At this stage, it is sometimes difficult to follow the author's line of argumentation.

2. It should be possible for the reader to follow the main ideas of the study without completely reading Goger et al. (2022). This particularly refers to the model setup. What is the vertical and horizontal grid spacing of the inner domain? What is the height of the lowest model level? Can you briefly summarize the main findings of the comparison of the REF run with the measurements during HEFEX? Why do the REF run and observations diverge after 1200?

3. Although the authors state in l. 324ff. that "we cannot assume that the local glacier boundary layer [...] is simulated realistically", RQs 2 and 3 and Section 4 focus on the effects of the overlying flow on the near-surface heat exchange in the local glacier boundary layer over HEF. I suggest giving a more comprehensive reasoning as to why analyzing the effects on the near-surface atmosphere is valid or focusing the manuscript on larger-scale interactions between the (breaking) gravity wave and other flows and leaving out near-surface processes.

4. Can you include a brief review on gravity waves (generation, overturning, breaking, hydraulic jump (l. 154)) and their influence on the isentropes and the turbulent kinetic energy as this is an important part of your study?

5. The introduction should present a comprehensive motivation for the conducted research. Can you clearly state the knowledge gap you address with this study? Your RQs 2 and 3 leave the impression that the study aims to investigate the effects of artificial surface modifications on the atmosphere. However, I think the authors performed the ice-removed runs as a method to gain more information on upstream influences on the atmosphere above HEF.

6. A separate method section that contains the definitions you use (e.g. Scorer parameter, up-valley wind index, advection formula, ...) can enhance the readability of the results section. Additionally include a description of the surface parameters used for ice and replaced ice surfaces in section 2.

7. In the discussion, add ideas on how to cope with the lack of near-surface process representation in your model. Would nesting with a finer-scale model, such as HICAR (Reynolds et al., 2023), close to the surface improve the results?

8. The discussion lacks a clear storyline. Please revise the structure of this section. Furthermore, a main part of the results section is dedicated to the surface energy exchange, but the results are not discussed.

**Specific Comments**

1. At the first occurrence, state that all times are in UTC and then remove the "UTC" throughout the rest of the manuscript.

2. l. 32f.: Mott et al. (2020) found the same over HEF

3. l. 51: High-resolution $\rightarrow$ decameter-resolution

4. l. 51: Introduce the "LES" abbreviation at the first occurrence

5. l. 51f.: Include Mott et al. (2019). They investigated the near-surface boundary layer over a perennial ice field using measurements and a high-resolution modelling setup.

6. l. 55: Can you detail more on why you are confident that the LES can resolve the "relevant mesoscale flow"? Here, it would help to talk about the results from Goger et al. (2022)

7. l. 69: What do you mean by "stronger due to the ice surfaces"?

8. Figure 1: color contours → contour lines and colors

9. l. 90: Can you give a number for ice melt during "extreme mass loss [...] in some recent years" to compare to the 1 m over the last 20 years?

10. l. 101: add a reference to Figure 1b in Goger et al. (2022)

11. l. 114: How do you define the upstream glacier surface replaced in NO_UP? What do you set as the "new" parameters describing the replaced surfaces in contrast to glacier ice?

12. l. 119: In addition, you use the current (ice-covered) DEM just replacing surface properties, right? So the topography of the replaced surfaces is also not representative of a melting ice cover under the influence of climate change.

13. l. 124ff.: This sentence fits in l. 106.

14. l. 127: More concise caption. Maybe: "Flow structure"?

15. l. 132ff.: Remove that sentence. Already stated above

16. l. 136ff.: That sentence would fit earlier in the manuscript when you talk about the selection of the period

17. all following figures: Can you make the color of the glacier outline consistent with Figure 1?

18. Figure 2: The $x$-labels of the color plots are overlapping.

19. Figure 2 caption: Height of the lowest model level?

20. l. 140: Height of the lowest model level?

21. Figure 3 caption: Indicate, that the cross sections are taken looking up valley along the black line in Fig. 1b.

22. interpretation of figure 3: If you talk about stability, you mean "static" stability, right? Could you note that when you talk about stability inferred from the isentropes?

23. interpretation of figure 3: Include the Scorer parameter plots in figure 3 and discuss the parameter in the course of figure 3 in connection with the gravity waves to improve the storyline.

24. l. 145: Can you support your note on "weakening of the cross-glacier flow" with values? I find it hard to see in Fig. 2e,f.

25. l. 150f.: repetition of the previous sentence

26. l. 157: $\Delta\mathrm{TKE} = -5\,\mathrm{m}^2\,\mathrm{s}^{-2}$

27. l. 171: Indicate the upstream point in figure 1

28. l. 176: lower $\rightarrow$ weaker

29. l. 180: Fig. 4e?

30. l. 179ff.: Briefly note that the near-surface stratification is often different from further aloft.

31. l. 180f.: Do you have a hypothesis as to whether the jet height is different?

32. l. 183: maxima $\rightarrow$ extrema, large decrease $\rightarrow$ peak

33. l. 185: within $\approx 600\,\mathrm{m}$ above the surface

34. Figure 4: Set the upper limit for the $y$-axes to $4000\,\mathrm{m}$ like figure 3. Consider creating separate subplots for the wind direction to enhance the readability of the plots. i,j,k,l) Sc $\rightarrow l^2$

35. l. 186: potential temperature difference close to the surface

36. l. 187f.: The REF simulation at the upstream location also shows a $\approx 100\,\mathrm{m}$ deep near-neutral or slightly unstable layer adjacent to the surface.

37. l. 192f.: Check sentence structure

38. l. 195: differences in the potential temperature profiles close to the surface

39. l. 199: the flow changes direction at $4000\,\mathrm{m}$, which is above crest height

40. Figure 6: Could you indicate the cross-glacier and along-glacier flow directions in panel a? Why did you leave the thermally-driven regime out in panel b? Spell out "UWI" in the caption and add a reference to (2).

41. l. 205f.: Where do you identify the neutral layer and the inversion in fig. 5e?

42. l. 210: Upper part of HEF

43. l. $204 - 215$: challenging to follow

44. l. 233ff.: At which location and height do you extract the upstream wind direction? The same as before? Can you indicate that in fig. 1?

45. l. 235: add a reference to figure 1 in Whiteman and Doran (1993)

46. l. 250f.: Is the model capable of resolving thermally-driven winds so you can say their influence is negligible? Please add a brief comment on that.

47. l. 257ff.: Split this sentence and be more precise about the indirect dependence of the heat fluxes on the wind direction (via air temperature) in the Monin–Obukhov formulation. Consider presenting the formula and indicate how you diagnose sensible heat fluxes from your model output.

48. section 4.1: Make clear when you refer to positive in contrast to negative heat fluxes. I suggest using the terms more/less pronounced or stronger/weaker when you relate fluxes of the same sign instead of higher/lower. Indicate, when you compare fluxes of different signs (mainly in the presentation of fig. 8).

49. l. 262f.: negative heat flux corresponds to the transport of heat from the atmosphere into the ice (atmospheric notation)

50. l. 264: Which pattern are you referring to?

51. l. 275: SH fluxes: be consistent with abbreviations

52. l. 274−281: Consider moving fig. 8 and this paragraph to the supplements and just give a summary in the main study. The results of NO_GL do not seem surprising and there is already a lot of information.

53. l. Figure 7 interpretation: Can you go into more detail about different sensible heat flux magnitudes on HEF and how they relate to the local wind direction? That would help to highlight the importance of the local wind direction on the surface energy exchange and reinforce the manuscript.

54. Figure 7: $x$-labels overlap

55. Figure 7 and 8: Consider focusing the color bar extent to HEF to make the differences on the glacier more apparent. Include a sentence about the wind arrows in the captions.

56. Figure 8 right column: outlines of replaced ice surfaces missing

57. Figure 9: Consider splitting this figure into two figures: The first containing the left column and the second the right column. The subfigures in the right column miss $y$-axis labels. Furthermore, referring to the right column, in l. 298 − 308, you are mostly analyzing the heating effects at the glacier surface. Consider a simpler time-series diagram with just the surface values or discuss the vertical structure in more detail.

58. l. 282: Heat Advection and Heat Budget? Consider splitting this into two subsections.

59. l. 283: At which height are you extracting the data from the model?

60. l. $285 - 295$: Please revise this paragraph. I can not follow your presentation.

61. l. 296: What do you mean by vertical heat budget? The temperature tendency equation? add original citation (Wyngaard, 2010)

62. l. 297: what terms have you neglected and why?

63. l. 300: a.g.l.

64. l. 300: what do you mean by wavy structure? Spatially or temporally?

65. l. 302: but there are periods of $\frac{\partial \theta}{\partial t} < 0$ at the surface between $0945 - 1000$ and $1145 - 1200$

66. l. $309 - 321$: I suggest focusing on the effects on HEF.

67. l. 335: you noted earlier that REF is not reliable after 1200

68. l. 345f.: missing end of sentence

69. l. 346: Which forces? Refer to Whiteman and Doran (1993)

70. l. 350: A large part of the results section is dedicated to the surface energy exchange, but the results are not discussed here

71. l. 380: The transition between the summarizing sentences and the outlook is abrupt.

**References**

Brigitta Goger, Ivana Stiperski, Lindsey Nicholson, and Tobias Sauter. Large-eddy simulations of the atmospheric boundary layer over an alpine glacier: Impact of synoptic flow direction and governing processes. *Quarterly Journal of the Royal Meteorological Society*, 148:1319–1343, 4 2022. ISSN 0035-9009. doi: 10.1002/qj.4263.

Rebecca Mott, Andreas Wolf, Maximilian Kehl, Harald Kunstmann, Michael Warscher, and Thomas Grünewald. Avalanches and micrometeorology driving mass and energy balance of the lowest perennial ice field of the alps: a case study. *The Cryosphere*, 13:1247–1265, 4 2019. ISSN 1994-0424. doi: 10.5194/tc-13-1247-2019.

Rebecca Mott, Ivana Stiperski, and Lindsey Nicholson. Spatio-temporal flow variations driving heat exchange processes at a mountain glacier. *The Cryosphere*, 14:4699–4718, 12 2020. ISSN 1994-0424. doi: 10.5194/tc-14-4699-2020.

Dylan Reynolds, Ethan Gutmann, Bert Kruyt, Michael Haugeneder, Tobias Jonas, Franziska Gerber, Michael Lehning, and Rebecca Mott. The high-resolution intermediate complexity atmospheric research (hicar v1.1) model enables fast dynamic downscaling to the hectometer scale. *Geoscientific Model Development*, 16:5049–5068, 9 2023. ISSN 1991-9603. doi: 10.5194/gmd-16-5049-2023.

C. David Whiteman and J. Christopher Doran. The relationship between overlying synoptic-scale flows and winds within a valley. *Journal of Applied Meteorology*, 32:1669–1682, 11 1993. ISSN 0894-8763. doi: 10.1175/1520-0450(1993)032¡1669:TRBOSS¿2.0.CO;2.

John C. Wyngaard. *Turbulence in the Atmosphere*. Cambridge University Press, 1 2010. ISBN 9780521887694. doi: 10.1017/CBO9780511840524.

---

## Referee Comment (RC2)

**Review of "Investigating the influence of changing ice surfaces on gravity wave formation and glacier boundary-layer flow with large-eddy simulations"**

Cole Lord-May
`clordmay@eoas.ubc.ca`

Building upon a previous study, the authors present three 6 h large-eddy simulations of glacier-atmosphere interactions above Hintereisferner (HEF). In their case studies, ice surfaces are replaced with bare rock to better understand how stabilizing conditions at a larger *glacier-system* scale impact smaller outlet glaciers. The authors find that upstream glaciers act as a strong control on the outlet glacier, altering local gravity wave breakup, wind patterns, and spatiotemporal heat flux distributions.

The manuscript serves as a great numerical complement to the hypotheses laid out by Conway et al. [2021], and explores questions I've often asked when standing on the outlet glaciers of large icefields. The authors present some interesting insights into the role of upstream glaciers on local conditions, and in turn, on the future of glacier-scale surface energy balance modelling. While implications of their results are very compelling, the presentation could be improved to better sell the results themselves. Throughout, I suggest a better quantification of the results, as many of the conclusions drawn rely on the reader's ability to visually differentiate between spatial maps. Additionally, I'm wary of the use of WRF in resolving the near-surface glacier boundary layer, so a clearer statement of assumptions and exploration of limitations is needed to trust the model outputs.

Please contact me if you have any questions. I'm happy to provide the `.tex` file if it makes the response easier.

**General comments**

G1 A careful re-read for spelling and syntax throughout would improve the manuscript.

G2 Answering the second and third research questions hinges on WRF accurately resolving the stable boundary layer above the glacier surface. As stated in L324, this is a questionable assumption (although L55 suggests otherwise). As the authors found in Goger et al. [2022], the sensible heat flux was overestimated by roughly $2\times$. Although their grid is more coarse, Draeger et al. [2024] found an underestimation of sensible heat fluxes, but also found that the results varied significantly depending on the model parameterization scheme chosen. As accurate modelling of the glacier boundary layer is paramount to the results presented, the manuscript would benefit greatly from a systematic enumeration of assumptions and limitations, and a clear argument of why the model outputs are to be trusted. Many of my specific comments throughout are related to this point.

G3 Language like "deglaciation" or "removal of ice surfaces" is used to explain the case studies. How is this done? Is it a removal or a replacement? If the former, more explanation of how the underlying topography is inferred would be needed. If the latter, I would suggest changing the terminology and stating all of the ways in which this affects the WRF simulation (roughness lengths, temperature boundary condition, etc.).

G4 How does the "no glaciers" experiment answers the questions asked? I think many of these results could go into the supplementary material, as they did not vary substantially from the "HEF-only" simulations (except on HEF). Might it be more illustrative to run the simulations with the upstream glaciers while removing HEF? This would also better simulate "realistic" deglaciation.

G5 Gravity wave "formation" is used throughout, including the title. Is gravity wave break-up not what is being simulated?

G6 How representative is this 6h period of the typical flow conditions observed over HEF? This would help provide a take-away message that is more generalizable to other glaciers/study sites.

G7 The manuscript would benefit from a more holistic introduction to gravity waves. A schematic would make the interpretations of certain figures easier to follow. This should include a discussion of how WRF resolves gravity waves at the given scales. Stull [2015], p.761 highlights that the model grid spacing is not the model resolution, and wavelengths smaller than roughly $7\Delta x$ are often filtered out for numerical stability. What wavelengths do you expect to see, and how well does WRF resolve these wavelengths?

G8 I find it hard to substantiate the conclusion that one must, in general, view local flow dynamics in the context of a larger system given that this experiment was only 6h (especially without knowing how prevalent strong NW synoptic winds are at this location). While I know these simulations are expensive, I feel it would be illustrative to run a no-upstream-glaciers simulation under weaker, or otherwise different, synoptic forcing if this is the intended message.

G9 Can you comment more on the cause of the deviations between simulations and observations after 12:00? How do you argue that your experiment simulations are still valid at 12:00? In Fig 7, 8, and 10, the most pronounced changes (apart from the missing ice surfaces) seem to occur at the bottom of the domain, far from the removed ice surfaces. Is this physical? How far into the next valley do we see these effects?

G10 I would prefer all equations to be included in the methods section. In general, I would encourage a slight restructing so that the results presented can be better anticipated from the outset.

**Specific comments**

- I find that NO_UP and NO_GL read more like filenames than experiments, likely due to the underscore.

- L13: I do not think this study shows that a glacier tongue is *never* isolated from the surrounding glacier environment. (G8)

- L55: Can you argue this point more clearly? The scale difference between mesoscale and glacier-scale flow is at least a couple orders of magnitude. (G2)

- L105: Would you use a different module if the BL was not turbulent?

- L107: Three hours of spin-up time seems relatively low. Draeger et al. [2024] had a 24h spin-up time, and Liu et al. [2023] showed that the choice of spin-up time depends on process being modelled. To that end, Sun et al. [2014] highlights the importance of spin-up time in convective models. How are you certain that three hours is sufficient in the simulations without the stabilizing glacier surfaces? (G9)

- L113 (and elsewhere): "Sensitivity study" reads more like testing different parameter regimes than removing entire glaciers.

- L130: Non-stationarity meaning changing gradually throughout the day? Or referring to turbulent stationarity? If the latter, this also needs to be mentioned when introducing M-O theory.

- Figure 2 (and elsewhere): Here you present time as rows, and in other cases you present time as columns. Choosing one would be preferable. Additionally, axis labels here (and elsewhere) overlap.

- L157: $\Delta$ TKE?

- L163: Do you mean "shooting" downslope flow as classified by Mahrt [1982]? If so, please clarify this interpretation.

- Figure 4 a-d (and discussion): If the lowest level shown here is $3\,\mathrm{m}$ above the surface, then I am rather surprised that REF and NO_GL show a temperature difference of only $2\,\mathrm{K}$. How do you explain the increase in temperature toward the surface in the lowest levels of REF? What does "mostly neutral" mean? (G2)

- Figure 4 e-h (and discussion): These subplots are visually dense. Some of these profiles look quite surprising – Are there two velocity maxima in the NO_UP and NO_GL simulations? What is meant by L198 "chaotic behaviour"? (G2)

- Figure 4 i-l (and discussion): I do not see how the Scorer parameter profiles presented show conditions favourable for gravity waves. In the provided references [e.g. Parmhed et al., 2004], the Scorer parameter is larger near the surface and then decreases, and is more than an order of magnitude larger than the near-surface values presented here. The Scorer profiles here look like they're being significantly affected by the division by a small $U$. Additionally, how are these gradients calculated? (G2)

- L156-160: This presentation is a bit hard to follow. I would argue that the bulk method is agnostic to wind direction. That is, one does not check the wind direction to pick the temperature to use in the model, but rather uses whatever temperature is measured (which *is* likely different under different flow regimes, but then this is an implicit dependence, not an explicit one). Moreover, I'm a bit unsure why the bulk methods are being introduced here.

- Equation 1: Brackets aren't tall enough.

- L174: $N = N(z)$ is correct, but a bit misleading as it hides the $\frac{\partial \theta}{\partial z}$ dependence.

- Figure 5: I think units of m would be preferable along the slope.

- Figure 6: How do A and B relate to each other? It appears from A that the REF observations have the same variability (-0.5 to 1) as the NO_UP and NO_GL. Yet in B, The REF observations have a reduced range of observed wind directions relative to NO_UP and NO_GL.

- Equation 2: Typset this equation using "upright" cos and brackets of appropriate height. I also prefer UWI and wdir to be upright and not slanted here and in the text. The conversion to radians is implied. So I prefer,

$$\mathrm{UWI} = \cos|\mathrm{wdir} - \phi|,$$

or similar. That said, if the UWI is only computed at one point along the glacier, I feel that wind direction alone (perhaps oriented such that 0 is upglacier) is sufficient, and simpler. It seems the utility of this metric arises when comparing observations from multiple locations where "upslope" might be different.

- L223: Worthwhile to introduce the negative case. Perhaps more clear to state, say, $|\mathrm{UWI}| < 0.5$ indicates cross-glacier flow.

- L234: Why use upstream location and not wind direction from a higher level in the same column?

- Section 3.3: The latter half reads as a discussion and not a presentation of results.

- L276-281: As you say, none of these results are particularly surprising. (G4)

- Figures 7 and 8: The choice of colorbar scales makes interpretation challenging. I would be far more interested to see the differences in sensible heat flux into the glaciers. A colorbar with a nonlinear scaling might help here. Overlapping axis labels. Inconsistent sensible heat flux vs. "SH" (here and in text).

- Section 4.2: Please clarify the intended message of this section. The beginning sentences do not seem related to the section title, directly. It seems the focus is on advection of heat and not momentum?

- L290-292: What would be the foundation of this assumption?

- Equation 3: This is not a complete heat budget. Please explain which terms are omitted and why. What length scales are the derivatives taken over? ADV and vHFD appear upright in the text, so should appear upright here. That said, they are only used this once so don't need to be abbreviated.

- Figure 9: I prefer the colorbar only over the subpanels where it is relevant. (d-f) I feel would be better presented as components of the budget. Is the budget closed? I can't tell if this is the LHS or RHS of eqn. 3 at present. (c) Downglacier advection? Or total horizontal advection?

- Figure 10: Similar to before, is the message here related to the temperature differences over the whole domain, or the temperature differences on the glacier? The colorbars could show this better, if the latter. Panel d is not needed.

- L313: Why do you trust this assessment of $2\,\mathrm{m}$ temperature given the criticisms of M-O theory applied to katabatics [e.g. Grisogono et al., 2007, and references therein]. (G2)

- L330: What are dynamical aspects?

- L346: Earlier what?

- L355: If this is an intended take-away message, it would be good to (1) make this a more clear objective, and (2) quantify these effects more clearly. How did you decide on 5 km? The Columbia Icefield is the study of Conway et al. [2021] is very large. Do you expect the size of the icefield to play a role? Do you expect this to be true in all synoptic conditions or just some? (G6/G8)

**References**

J. P. Conway, W. D. Helgason, J. W. Pomeroy, and J. E. Sicart. Icefield Breezes: Mesoscale Diurnal Circulation in the Atmospheric Boundary Layer Over an Outlet of the Columbia Icefield, Canadian Rockies. *Journal of Geophysical Research (Atmospheres)*, 126(6):e34225, Mar. 2021. doi: 10.1029/2020JD034225.

C. Draeger, V. Radić, R. H. White, and M. A. Tessema. Evaluation of reanalysis data and dynamical downscaling for surface energy balance modeling at mountain glaciers in western canada. *The Cryosphere*, 18(1):17–42, 2024.

B. Goger, I. Stiperski, L. Nicholson, and T. Sauter. Large-eddy simulations of the atmospheric boundary layer over an alpine glacier: Impact of synoptic flow direction and governing processes. *Quarterly Journal of the Royal Meteorological Society*, 148(744):1319–1343, 2022.

B. Grisogono, L. Kraljević, and A. Jeričević. The low-level katabatic jet height versus monin–obukhov height. *Quarterly Journal of the Royal Meteorological Society: A journal of the atmospheric sciences, applied meteorology and physical oceanography*, 133(629):2133–2136, 2007.

Y. Liu, L. Zhuo, and D. Han. Developing spin-up time framework for wrf extreme precipitation simulations. *Journal of Hydrology*, 620:129443, 2023.

L. Mahrt. Momentum balance of gravity flows. *Journal of Atmospheric Sciences*, 39(12):2701–2711, 1982.

O. Parmhed, J. Oerlemans, and B. Grisogono. Describing surface fluxes in katabatic flow on breidamerkurjökull, iceland. *Quarterly Journal of the Royal Meteorological Society: A journal of the atmospheric sciences, applied meteorology and physical oceanography*, 130(598):1137–1151, 2004.

R. B. Stull. *Practical meteorology: an algebra-based survey of atmospheric science*. University of British Columbia, 2015.

J. Sun, M. Xue, J. W. Wilson, I. Zawadzki, S. P. Ballard, J. Onvlee-Hooimeyer, P. Joe, D. M. Barker, P.-W. Li, B. Golding, et al. Use of nwp for nowcasting convective precipitation: Recent progress and challenges. *Bulletin of the American Meteorological Society*, 95(3):409–426, 2014.

---

## Author Comment (AC1)

**Response to Michael Haugeneder (referee #1)**
**Investigating the influence of changing ice surfaces on gravity wave formation and glacier boundary-layer flow with large-eddy simulations**

Brigitta Goger, Lindsey Nicholson, Matthis Ouy, and Ivana Stiperski

November 21, 2024

Dear Michael Haugeneder,

We would like to thank you for the thorough evaluation of our manuscript. Below we address our detailed responses to all the comments. In this response document we try to clarify and address each of your suggestions, comments, and questions by you. Therefore we have copied the comments in blue boxes and have addressed them one by one. In the response we use italic fonts to quote text from the revised manuscript. In some cases, we explain why we didn't concur with your suggestion. In addition to the revised manuscript, we have uploaded a version of the manuscript with highlighted track changes that indicate where the manuscript has changed (**red**=changes made in the revised manuscript).

Best regards,
Brigitta Goger, Lindsey Nicholson, Matthis Ouy, and Ivana Stiperski

> The authors present a case study of large-eddy simulations of the atmosphere above the highly heterogeneous high-mountain terrain surrounding Hintereisferner (HEF). They compare a reference simulation featuring realistic ice surfaces with two sensitivity anal- yses comprising different surface properties. In one run, they only replace the upstream glacier surfaces, while in the other sensitivity run, all glacier surfaces are replaced by bare ground. With these different setups, the authors aim at investigating the influence of upstream surface properties on the flow dynamics over the HEF ultimately affecting the heat fluxes in the near-ice atmosphere at HEF.
>
> Their work contributes significantly to understanding the larger-scale (and non-local) influences on the local flow field. Comparing runs with modified surface properties to verified "real-world" runs offers promising insights advancing the current understanding of process interactions at a wide range of scales. However, I suggest revising the presen- tation of the results to facilitate better understanding.
>
> If the authors have any questions, please don't hesitate to ask.

We thank you for the encouraging comments, especially for the detailed inquiries on the model's performance, which will lead to an improvement of the revised manuscript.

**Major comments**

> 1. language and spelling: The manuscript will strongly profit from a thorough revision of the language including syntax and spelling. At this stage, it is sometimes difficult to follow the author's line of argumentation.

Thank you for the comment. We re-read the revised manuscript thoroughly and hope that the readability improved and no typos are present anymore. Furthermore, we modified the figures according to your suggestions.

> 2. It should be possible for the reader to follow the main ideas of the study without completely reading Goger et al. (2022). This particularly refers to the model setup. What is the vertical and horizontal grid spacing of the inner domain? What is the height of the lowest model level? Can you briefly summarize the main findings of the comparison of the REF run with the measurements during HEFEX? Why do the REF run and observations diverge after 1200?

- We agree that the current manuscript should be readable without knowing the contents of Goger et al. (2022). We already mention the horizontal grid spacing and the vertical grid spacing in Section 2.2:
  *"We use a nested set-up consisting of four domains, where the outermost domain spans Europe with $\Delta x$=6 km and receives ERA5 reanalyses (Hersbach et al., 2020) as boundary and initial conditions. We subsequently nest down over $\Delta x$=1 km, and $\Delta x$=240 m to the innermost domain at $\Delta x$=48 m (Goger et al., 2022, their Figure 1b)."*
  and furthermore
  *"In this study, we only show the output domain 4 for our analysis, and any mentioned numerical data will stem from this domain at $\Delta x$=48 m."*

- We mentioned the lowest model level in the results, but we agree that we have to introduce it already in Section 2.2:
  *"The lowest model level in the innermost domain is located at 7 m, resulting in the lowest model half-level height of $z = 3.5\,m$."*

- Unfortunately, we have to summarize the results from Goger et al. (2022) in a concise way to avoid repeating ourselves, and we give an overview of the major results of the model run in comparison with the HEFEX observations at the beginning of Section 3:
  *"The reference simulation (REF) is a real-case simulation of the glacier boundary layer from August 17, 2018. Under North-Westerly synoptic influence, a gravity wave formed over the North-Western ridge close to HEF, leading to a continuous disturbance of the glacier boundary layer. The case study day was dominated by cross-glacier flow and high values of non-stationarity (Mahrt, 1998) of the sensible heat flux during gravity-wave breaking episodes and a strong mesoscale influence on the glacier boundary layer. Details on the simulation and further results can be found in Goger et al. (2022), their "NW day"."*

- In general, in real-case LES there is always the so-called 'scale separation' problem, meaning that large-scale flows and smaller scales interact with each other (Schemann et al., 2020). This is also evident in our simulation, especially due to the strong synoptic forcing, and the large-scale flow starts to dominate over smaller scale features due to misrepresented scale interactions (e.g., as discussed in Goger et al. (2022) with the 'too strong' gravity wave or in (Umek et al., 2021) with foehn-cold air pool interactions). To overcome this deviation of results after 12:00, we would have to re-start (re-initialize) the model again with a new analysis field. However, observations suggest that the situation over the glacier does not change drastically therefore that the restarted simulations would not bring many new insights into the topic, therefore we decided to stay with the 6 hour period for our analysis. We added the following sentence to the beginning of Section 3:
  *"The REF starts to deviate from the observations after 12:00 UTC (Goger et al., 2022, their Fig. 2) due to the scale separation problem in LES (Schemann et al., 2020), but observations suggest that the situation over the glacier does not change drastically after 12:00 UTC. This period is relatively short, but since observations suggest that the situation over the glacier does not change drastically from 12:00 until sunset, an extended analysis does not bring new insights. "*

- In general, we extended Section 2 and now also discuss the assumptions and shortcomings of our current model setup.

> 3. Although the authors state in l. 324ff. that "we cannot assume that the local glacier boundary layer [...] is simulated realistically", RQs 2 and 3 and Section 4 focus on the effects of the overlying flow on the near-surface heat exchange in the local glacier boundary layer over HEF. I suggest giving a more comprehensive reasoning as to why analyzing the effects on the near-surface atmosphere is valid or focusing the manuscript on larger-scale interactions between the (breaking) gravity wave and other flows and leaving out near-surface processes.

Generally speaking, we are aware of the shortcomings of our simulations, e.g., that MOST is likely invalid over complex terrain, and that scale interactions pose a challenge for high-resolution numerical models due to the scale separation problem (Schemann et al., 2020). However, we discuss these challenges both in the current manuscript and also in our previous publications with similar setups (Goger et al., 2022; Voordendag et al., 2024). Our major findings include that the absolute values of quantities (e.g., horizontal wind speed, 2 m temperature, etc), exhibit a certain bias, but we also showed that the relevant physical processes (e.g., dependence of the sensible heat flux on the wind speed, stationarity of sensible heat fluxes, horizontal advection patterns, and the heat budget) are represented well in the model. We have to emphasize that we now have a starting point with these simulations –

of course, we could simulate plenty more cases, but given the expensiveness of the simulations, we have to focus on single case studies. We think that this particular NW day is an excellent day to study the impact of large-scale flows on the glacier boundary layer, especially because we can expect that the gravity wave and associated processes are already resolved on the grid.

Furthermore, we have to differentiate between the local glacier boundary layer, i.e., a boundary layer dominated by katabatic flows, and the 'disturbed' boundary layer we noted in simulations and observations (Mott et al., 2020; Goger et al., 2022). The small-scale processes in stable boundary layers (with katabatic flows) would require much higher horizontal and vertical grid spacings to be resolved fully - Cuxart (2015) even suggests horizontal grid spacings of $1\,\mathrm{m}$, which is clearly not met by our simulations.

However, in our case study the local glacier boundary layer is not able to form due to the strong dominance of the gravity wave in the REF simulation (Goger et al., 2022). Therefore, we do not have a down-glacier katabatic flow (cross-glacier flow instead), and as the vertical profiles reveal in Fig. 4, rather a neutral boundary layer above the location glacier tongue. Therefore, we have a gravity-wave-dominated, almost neutral boundary layer above the glacier tongue, and we can expect from our model that surface-exchange processes under these conditions can be expected to be resolved correctly.

We, however, agree with the referee that the statement in the manuscript is misleading. We added one clarifying sentence to the beginning of Section 3:

*"Due to the aforementioned phenomena, the local glacier boundary layer is heavily disturbed both in simulations and observations, and no katabatic down-glacier flow is present."*

> 4. Can you include a brief review on gravity waves (generation, overturning, breaking, hydraulic jump (l. 154) and their influence on the isentropes and the turbulent kinetic energy as this is an important part of your study?

Thank you for this comment - we re-organized our introduction and dedicate a paragraph to gravity waves and also added additional literature references.

> 5. The introduction should present a comprehensive motivation for the conducted research. Can you clearly state the knowledge gap you address with this study? Your RQs 2 and 3 leave the impression that the study aims to investigate the effects of artificial surface modifications on the atmosphere. However, I think the authors performed the ice-removed runs as a method to gain more information on upstream influences on the atmosphere above HEF.

We wish to investigate the influence of the upstream glaciers on the atmospheric flow structure and gravity wave formation, since the NW day case study of Goger et al. (2022) showed that under NW flow conditions, the local glacier boundary layer is eroded. Since previous literature (Turton et al., 2018; Jonassen et al., 2014) suggests that ice surface have an impact on the atmospheric static stability aloft, we want to investigate the role of removing these upstream ice surfaces on the flow structure over HEF. We want to stress that these numerical simulations are, although semi-idealized, the only way to investigate this question.

Since our case study is under the influence of strong (synoptic) North-Westerly flow, we cannot isolate the 'upstream location' and the 'atmosphere above HEF' from each other and rather have to view them as a connected system (this was also highlighted in the final conclusion on a 'system of glaciers' instead of isolated glacier tongues).

We re-formulated the research questions, though, and hope now to address the open questions and knowledge gaps better.

> 6. A separate method section that contains the definitions you use (e.g. Scorer parameter, up-valley wind index, advection formula, ...) can enhance the readability of the results section. Additionally include a description of the surface parameters used for ice and replaced ice surfaces in section 2.

We removed the equations from the Section 2 and added a subsection called "analyses performed". We also added a Table (Tab. 1) about the changes in the surface parameters and an accompanying description text to the manuscript (Section 2.2).

| Land-use category | Albedo (%) | Moisture availability (%) | Emissivity (% at $9\,\mu$m) | $z_0$ (cm) | Thermal inertia ($1\,\mathrm{W\,m^{-2}\,k^{-1}\,s^{1/2}}$) |
|---|---|---|---|---|---|
| Snow or ice | 41.5 | 95 | 96.1 | 5 | 418 |
| Bare rock | 16.9 | 2 | 96.5 | 10 | 2948 |

Table 1: Surface parameters from the CORINE dataset for the two land-use categories 'snow or ice' and 'bare rock' for the summer season after Pineda et al. (2004).

> 7. In the discussion, add ideas on how to cope with the lack of near-surface process representation in your model. Would nesting with a finer-scale model, such as HICAR (Reynolds et al., 2023), close to the surface improve the results?

We discuss the challenge in simulating near-surface processes throughout the manuscript and also mention the challenges of the model in the first paragraph of the discussion, where we cite literature on the very setup and the discussion there, so we cannot repeat already published work one-by-one. We are aware that there will be always a certain misrepresentation of surface processes in the model, but we also want to stress that we investigate a case study day where we simulate strong dynamical forcing - so small-scale processes are not so important as in, i.e., katabatically-driven cases.

To our current knowledge, HICAR's target resolution is $50\,$m according to Reynolds et al. (2023, 2024). Our innermost LES domain (the analysis domain for this current manuscript) has a horizontal grid spacing of $48\,$m, therefore, a HICAR simulation would only bring advantages if the horizontal resolution were finer than $48\,$m. However, we can think about running simulations with intermediate complexity models as HICAR, especially to perform seasonal runs to explore further how often strong cross-glacier flows occur over HEF or any other glacier or icefield worldwide. This would also help answering the questions on representativeness raised by Cole Lord-May (referee #2). We added the following paragraph to the discussion:

*"A final open question to discuss is how representative our 6 hours of simulation are for HEF and its surroundings. Currently, we can only compare to a wind climatology at HEF compiled by Obleitner (1994), and they found a significant Northerly gradient wind influence on the South-facing slope of the valley, the same wind direction as in our case study. Furthermore, Mott et al. (2020) noted in the HEFEX campaign that in 20% of their wind observations the katabatic flow was 'disturbed' and the glacier boundary layer was eroded. Therefore, we can assume that the described situation of strong North-Westerly winds and gravity waves eroding the glacier boundary layer is not a single occurrence. Still, an updated wind climatology over HEF is necessary to quantify these events. Furthermore, applying an intermediate complexity model such as HICAR (Reynolds et al., 2023, 2024) to the region for entire seasons would shed more light on the typical wind patterns over HEF while being computationally cheaper than a full-physics LES."*

> 8. The discussion lacks a clear storyline. Please revise the structure of this section. Furthermore, a main part of the results section is dedicated to the surface energy exchange, but the results are not discussed.

We agree, and we revised our discussion section thoroughly. Pasting everything here would be too much, but we would like to direct you to our tracked changes manuscript to see the substantial changes we made on the discussion.

**Specific comments**

> 1. At the first occurrence, state that all times are in UTC and then remove the "UTC" throughout the rest of the manuscript.

Thank you. We added a sentence at the first occurence of UTC:
*"All further time information in this publication refers to UTC, so we will omit 'UTC' at all further occurrences."*

> 2. l. 32f.: Mott et al. (2020) found the same over HEF

The referee is right, but we discuss the findings by Mott et al. (2020) in the following paragraph in much more detail (as one of our major motivations for this study), therefore we decided to cite them at this later occasion.

> 3. l. 51: High-resolution → decameter-resolution

Changed the sentence to:
*High-resolution large-eddy simulations (LES) at decameter grid spacings have emerged [...]*

> 4. l. 51: Introduce the "LES" abbreviation at the first occurrence

Done.

> 5. l. 51f.: Include Mott et al. (2019). They investigated the near-surface boundary layer over a perennial ice field using measurements and a high-resolution modelling setup.

Thank you for the reference, we added it.

> 6. l. 55: Can you detail more on why you are confident that the LES can resolve the "relevant mesoscale flow"? Here, it would help to talk about the results from Goger et al. (2022).

We changed the sentence to:
*With a horizontal mesh size of 48 m, the topography governing the mesoscale flow evolution is already sufficiently resolved in the model for the successful simulaiton of structures and wind patterns on the glacier for both summer and winter [...]*

> 7. l. 69: What do you mean by "stronger due to the ice surfaces"?

We changed the sentence to:
*[...] downslope windstorms are stronger due to the stabilizing effect of the ice surfaces [...]*

> 8. Figure 1: color contours → contour lines and colors

Changed accordingly.

> 9. l. 90: Can you give a number for ice melt during "extreme mass loss [...] in some recent years" to compare to the 1 m over the last 20 years?

Thank you, the sentence now reads
*"While it loses around 1 m ice thickness per year over the last 20 years (Piermattei et al., 2024), extreme mass loss has been observed in some recent years (-3319 kg m$^{-2}$ in 2022, 3.2 times higher than the long-term mean for 1991-2020, Voordendag et al., 2023)."*

> 10. l. 101: add a reference to Figure 1b in Goger et al. (2022)

Done.

> 11. l. 114: How do you define the upstream glacier surface replaced in NO_ UP? What do you set as the "new" parameters describing the replaced surfaces in contrast to glacier ice?

Thank you for this remark, we added an additional explanatory sentence:
*The surfaces of the missing glaciers in the sensitivity simulations are replaced with the land-use category of the surroundings, namely bare rock.*

> 12. l. 119: In addition, you use the current (ice-covered) DEM just replacing surface properties, right? So the topography of the replaced surfaces is also not representative of a melting ice cover under the influence of climate change.

We did not change anything on the used DEM, but at a horizontal grids spacing of 48 m it cannot be expected that ice-covered surfaces were resolved in the model in the REF runs. We agree with the referee, this is not representative for a melting ice cap under climate change, which was not our intention of the study. As mentioned above, we added extra information to the manuscript:
*"The surfaces of the missing glaciers ('snow or ice') in the sensitivity simulations are replaced with the land-use*

*category of the surroundings, namely 'bare rock', while the topography remains the same."*

**13. l. 124ff.: This sentence fits in l. 106.**

Thank you, we moved the sentence upwards as suggested.

**14. l. 127: More concise caption. Maybe: "Flow structure"?**

Changed to *"Simulated flow structure with [...]"*

**15. l. 132ff.: Remove that sentence. Already stated above**

Removed the sentence.

**16. l. 136ff.: That sentence would fit earlier in the manuscript when you talk about the selection of the period**

We agree and moved the sentence to Section "Numerical Model".

**17. all following figures: Can you make the color of the glacier outline consistent with Figure 1?**

Yes. Now, all glacier outlines have the same colour as in Fig. 1.

**18. Figure 2: The x-labels of the color plots are overlapping.**

We changed this (and in all the follow-up figures with the same issue as well).

**19. Figure 2 caption: Height of the lowest model level?**

Added.

**20. l. 140: Height of the lowest model level?**

Added to the caption of Fig. 2.

**21. Figure 3 caption: Indicate, that the cross sections are taken looking up valley along the black line in Fig. 1b.**

Added.

**22. interpretation of figure 3: If you talk about stability, you mean "static" stability, right? Could you note that when you talk about stability inferred from the isentropes?**

Thank you, we added "near-surface static stability".

**23. interpretation of figure 3: Include the Scorer parameter plots in figure 3 and discuss the parameter in the course of figure 3 in connection with the gravity waves to improve the storyline.**

We indeed thought about this, but we think that the Scorer parameter (and the equation) fits better with the vertical profiles, since they directly influence its calculation.

**24. l. 145: Can you support your note on "weakening of the cross-glacier flow" with values? I find it hard to see in Fig. 2e,f.**

We noted that we referenced the wrong figure sub-panels, this was corrected. Furthermore, we re-wrote the sentence more clearly to
*"In the NO_UP and NO_GL simulations, however, we note the weakening of the cross-glacier flow, visible in reduced wind speeds after 08:00 UTC (wind arrows in Fig 3g,h)"*

**25. l. 150f.: repetition of the previous sentence**

True, we removed the sentence from the previous paragraph, and changed the sentence to:
*To better understand the changes in gravity wave formation in the NO_UP and NO_GL simulations compared to REF, we examine the vertical structure of the upstream flow conditions in the next paragraphs.*

**26. l. 157: $\Delta$ TKE = -5 m$^2$ s$^{-2}$**

Added.

**27. l. 171: Indicate the upstream point in figure 1**

The upstream point is indicated, it's the plus sign in Fig. 1. We mention it now again in the caption of Fig. 4.

**28. l. 176: lower $\rightarrow$ weaker**

We decided to go with "reduced stability".

**29. l. 180: Fig. 4e?**

True, thank you.

**30. l. 179ff.: Briefly note that the near-surface stratification is often different from further aloft.**

Changed the sentence to:
In all simulations at 06:00 UTC, the potential temperature profile at HEF tongue reveals a very shallow part with strong gradients, capped by a mixed layer influenced by gravity wave breaking, and topped by stable background stratification. So we see a three-layer structure on the vertical profiles, very common in Alpine terrain (Weigel et al., 2006).

**31. l. 180f.: Do you have a hypothesis as to whether the jet height is different?**

The jet height in stable boundary layer flows is dependent on the sensible heat flux - as the sensible heat fluxes are weaker in the NO_UP simulation, and even reverse sign in the NO_GL simulation, we assume that this change in the surface forcing also impacts the jet height.

**32. l. 183: maxima $\rightarrow$ extrema, large decrease $\rightarrow$ peak**

We think it's better to keep "maxima" instead of "extrema", since we did not perform a statistical extreme value analysis on our simulation data. Changed large decrease to peak.

**33. l. 185: within $\approx$ 600 m above the surface**

Added.

**34. Figure 4: Set the upper limit for the y-axes to 4000 m like figure 3. Consider creating separate subplots for the wind direction to enhance the readability of the plots. i,j,k,l) Sc $\rightarrow$ l2**

We need to set the ylim to 5500 m for the analysis of changes in the Scorer parameter, pivotal to explain the formation of gravity waves over the upstream glaciers.
We agree that especially the panels on wind speed and direction are messy. Therefore, we changed the opacity of the wind direction points to increase the figure's readability.

**35. l. 186: potential temperature difference close to the surface**

Changed accordingly.

36. l. 187f.: The REF simulation at the upstream location also shows a ≈ 100 m deep near-neutral or slightly unstable layer adjacent to the surface.

We agree and think that this is mostly related to the gravity wave and the turbulent mixing it induced. We already discuss this in (Goger et al., 2022), but we will mention it here again.

37. l. 192f.: Check sentence structure

Re-wrote the sentence to:
*The wind direction over the glacier tongue now reveals a distinct up-valley flow in the NO_UP and NO_GL simulations (Fig. 4g), while the flow remains cross-glacier in the REF simulation. At the upstream location, the flow is North-Westerly for all simulations.*

38. l. 195: differences in the potential temperature profiles close to the surface

Changed.

39. l. 199: the flow changes direction at 4000 m, which is above crest height

We agree - we replaced "below crest height" with "below 4000 m".

40. Figure 6: Could you indicate the cross-glacier and along-glacier flow directions in panel a? Why did you leave the thermally-driven regime out in panel b? Spell out "UWI" in the caption and add a reference to (2).

We added all proposed regimes after Whiteman and Doran (1993) to the revised figure. We spelled out "UWI" in the caption as suggested.

41. l. 205f.: Where do you identify the neutral layer and the inversion in fig. 5e?

We revised this text heavily and this formulation does not appear anymore in the revised manuscript.

42. l. 210: Upper part of HEF

Done.

43. l. 204-215: challenging to follow

We agree and re-wrote the paragraph to:
*"In the REF simulation there is a strong stable boundary layer (SBL) at the upper parts of the glacier at 06:00 UTC (Fig. 3a). The SBL coincides with a down-glacier flow, while below 3000 m a.m.s.l., the flow weakens and with reduced stability and higher TKE values, related to the strong cross-glacier flow and the gravity wave present (Fig. 3). In NO_GL, the isentropes show that there is generally weaker stratification (Fig. 5e) in accordance with lower TKE values associated with the weaker gravity wave (Fig. 5i). Two hours later, REF still shows a SBL over the upper part of HEF, however, in NO_GL, the stratification over the missing glacier is continuously weakened. Furthermore, NO_GL exhibits higher TKE values than REF below 3000 m a.m.s.l., related to the earlier breaking cross-glacier gravity wave. The atmosphere above HEF is well-mixed in both REF and NO_GL at 10:00 and 12:00 UTC (Fig. 5c,d,g,h), and the SBL mostly dissipated in both simulations, while the vertical profiles in the NO_GL simulation even suggest a convective boundary layer at the glacier tongue (Fig. 4d)."*

44. l. 233ff.: At which location and height do you extract the upstream wind direction? The same as before? Can you indicate that in fig. 1?

Exactly, we use the upstream location indicated as a plus in Figure 1. We now mention it again in the caption of Fig. 6.

45. l. 235: add a reference to figure 1 in Whiteman and Doran (1993)

Thank you, done.

46. l. 250f.: Is the model capable of resolving thermally-driven winds so you can say their influence is negligible? Please add a brief comment on that.

Yes, it is very realistic that thermally-induced flows are resolved in our model, at least the plain-to-mountain circulation, up- or down-valley flows, and up- or down-slope flows. According to Wagner et al. (2014), ten grid points across a valley are necessary to resolve the relevant mountain boundary-layer features. This criterion is met for our simulations at $\Delta x = 48\,\text{m}$ spanning across the $\approx 3\,\text{km}$ wide glacier valley with 62 grid points.

47. l. 257ff.: Split this sentence and be more precise about the indirect dependence of the heat fluxes on the wind direction (via air temperature) in the Monin–Obukhov formulation. Consider presenting the formula and indicate how you diagnose sensible heat fluxes from your model output.

We agree that our argumentation was somewhat chaotic and changed the paragraph to
*"The surface sensible heat flux plays a pivotal role in the energy exchange over glaciers in the summer months. Over a melting glacier the surface temperature is constant at $0°\,C$, impacting the bulk formulation of the sensible heat flux (Stull, 1988, their equation 7.4.1d). As the surface temperature over a melting glacier is exactly $0°$, sensible heat fluxes strongly depend on the wind speed, we can expect that changing wind patterns drive changes in the sensible heat flux structure."*
And we moved the paragraph to the 'sensible heat fluxes' subsection.

48. section 4.1: Make clear when you refer to positive in contrast to negative heat fluxes. I suggest using the terms more/less pronounced or stronger/weaker when you relate fluxes of the same sign instead of higher/lower. Indicate, when you compare fluxes of different signs (mainly in the presentation of fig. 8).

Thank you for your suggestions, we checked throughout the manuscript and now use the atmospheric notation continuously.

49. l. 262f.: negative heat flux corresponds to the transport of heat from the atmosphere into the ice (atmospheric notation)

Changed accordingly.

50. l. 264: Which pattern are you referring to?

We changed the sentence to:
*The reduced sensible heat fluxes are present during the entire simulation time at the remaining ice surface in NO_UP.*

51. l. 275: SH fluxes: be consistent with abbreviations

Changed to "sensible heat fluxes".

52. l. 274-281: Consider moving fig. 8 and this paragraph to the supplements and just give a summary in the main study. The results of NO_ GL do not seem surprising and there is already a lot of information.

There is one major finding we would like to highlight with the NO_GL simulation, namely that it shows that HEF as in isolated glacier tongue has almost no impact on the atmosphere aloft, despite the different sign of the sensible heat flux. This highlights that under our particular NW day situation, the glacier is not able to maintain its microclimate and that the upstream glaciers have indeed a larger impact on HEF's boundary layer than HEF itself. We will thereore keep Figure 8 in the main text.

53. l. Figure 7 interpretation: Can you go into more detail about different sensible heat flux magnitudes on HEF and how they relate to the local wind direction? That would help to highlight the importance of the local wind direction on the surface energy exchange and reinforce the manuscript.

This is related to whether the origin of the flow is located over an ice surface or on (heated) bare rock. We touch upon this subject already in Goger et al. (2022), where we state that sensible heat fluxes are generally higher in the NW day (this case study) simulation than if the source were over an ice.

In our current manuscript, we especially note enhanced sensible heat fluxes in NO_UP compared to REF at the glacier tongue, for cross-glacier flow as well as for the up-glacier flow. We discuss the sensible heat fluxes here:

*"The reduced sensible heat fluxes are present during the entire simulation time at the remaining ice surface in NO_UP. Interestingly, the sensible heat flux difference between NO_UP and REF is very small at the upper part of the remaining glacier, where cross-glacier flow is present (Fig. 7h,k,i,l). The largest differences between REF and NO_UP are visible at 12:00, when the gravity wave broke and the strong up-glacier flow is present (Fig. 7i,l). It is not surprising the sensible heat fluxes over the missing ice surfaces change sign and are positive now which range up to $500\,W\,m^{-2}$ between REF and NO_UP (Fig. 7k)."*

54. Figure 7: x-labels overlap

We changed the Figure accordingly.

55. Figure 7 and 8: Consider focusing the color bar extent to HEF to make the differences on the glacier more apparent. Include a sentence about the wind arrows in the captions.

We added the information on the wind arrows. We removed the heat fluxes of the surrounding terrain and now only focus on the (missing) ice surfaces.

56. Figure 8 right column: outlines of replaced ice surfaces missing

Thank you. We added the missing ice surfaces outlines.

57. Figure 9: Consider splitting this figure into two figures: The first containing the left column and the second the right column. The subfigures in the right column miss y-axis labels. Furthermore, referring to the right column, in l. 298 - 308, you are mostly analyzing the heating effects at the glacier surface. Consider a simpler time-series diagram with just the surface values or discuss the vertical structure in more detail.

Thank you for this nice suggestion. We split the Figure in two parts, and added the time-averaged components of the total vertical heat budget to the new Fig. 10. We adjusted the accompaying text and discuss the behaviour of the heat budget cmponents now.

58. l. 282: Heat Advection and Heat Budget? Consider splitting this into two subsections.

We renamed the section and added the equation of the horizontal temperature advection as well.

59. l. 283: At which height are you extracting the data from the model?

From the lowest model level - we added a clarification.

60. l. 285 - 295: Please revise this paragraph. I can not follow your presentation.

We revised the paragraph accordingly.

61. l. 296: What do you mean by vertical heat budget? The temperature tendency equation? add original citation (Wyngaard, 2010)

Yes, we mean the temperature tendency equation after (Wyngaard, 2010). We added the reference.

62. l. 297: what terms have you neglected and why?

We neglect the radiative flux divergence as in (Goger et al., 2022), because we consider it small during daytime. We added a sentence on that in the revised manuscript.

63. l. 300: a.g.l.

Changed.

**64. l. 300: what do you mean by wavy structure? Spatially or temporally?**

Temporally - we clarified the sentence.

**65. l. 302: but there are periods of $\frac{\partial \theta}{\partial t} < 0$ at the surface between 0945 - 1000 and 1145 - 1200**

We agree and reformulated the sentence to
*"This coincides to the gravity wave breaking pattern, because the brief periods positive $\frac{\partial \theta}{\partial t}$ correspond to the gravity wave breaking in REF (cf. Fig. 3), and the glacier is under warming from 09:00 until 12:00 UTC."*

**66. l. 309 - 321: I suggest focusing on the effects on HEF.**

Thank you, but we think it is also relevant information to discuss the effect of the ice surfaces on their immediate surroundings.

**67. l. 335: you noted earlier that REF is not reliable after 1200**

We agree, but we want to mention anyway that there is no change in patterns in the sensitivity simulations - another argument to omit further analysis after 12:00 UTC.

**68. l. 345f.: missing end of sentence**

True - we removed the last "and earlier".

**69. l. 346: Which forces? Refer to Whiteman and Doran (1993)**

We added the reference.

**70. l. 350: A large part of the results section is dedicated to the surface energy exchange, but the results are not discussed here**

We agree and added a paragraph on the surface exchange to the discussion.

**71. l. 380: The transition between the summarizing sentences and the outlook is abrupt.**

We re-wrote the sentences to
*"The present study gave insight to the impact of ice surfaces on gravity wave formation and breaking, and their impact on near-surface processes over a glacier. This is only a single case study, and in the future, similar studies with different upstream conditions could be conducted."*

**References**

[revised manuscript text omitted]

---

## Author Comment (AC2)

**Response to Cole Lord-May (referee #2)**
**Investigating the influence of changing ice surfaces on gravity wave formation and glacier boundary-layer flow with large-eddy simulations**

Brigitta Goger, Lindsey Nicholson, Matthis Ouy, and Ivana Stiperski

November 21, 2024

Dear Cole Lord-May,

We would like to thank you for the thorough evaluation of our manuscript. Below we address our detailed responses to all the comments. In this response document we try to clarify and address each of your suggestions, comments, and questions by you. Therefore we have copied the comments in blue boxes and have addressed them one by one. In the response we use italic fonts to quote text from the revised manuscript. In some cases, we explain why we didn't concur with your suggestion. In addition to the revised manuscript, we have uploaded a version of the manuscript with highlighted track changes that indicate where the manuscript has changed (**red**=changes made in the revised manuscript).

Best regards,
Brigitta Goger, Lindsey Nicholson, Matthis Ouy, and Ivana Stiperski

> Building upon a previous study, the authors present three 6 h large-eddy simulations of glacier-atmosphere interactions above Hintereisferner (HEF). In their case studies, ice surfaces are replaced with bare rock to better understand how stabilizing conditions at a larger glacier-system scale impact smaller outlet glaciers. The authors find that upstream glaciers act as a strong control on the outlet glacier, altering local gravity wave breakup, wind patterns, and spatiotemporal heat flux distributions. The manuscript serves as a great numerical complement to the hypotheses laid out by Conway et al. (2021), and explores questions I've often asked when standing on the outlet glaciers of large icefields. The authors present some interesting insights into the role of upstream glaciers on local conditions, and in turn, on the future of glacier-scale surface energy balance modelling. While implications of their results are very compelling, the presentation could be improved to better sell the results themselves. Throughout, I suggest a better quantification of the results, as many of the conclusions drawn rely on the reader's ability to visually differentiate between spatial maps. Additionally, I'm wary of the use of WRF in resolving the near-surface glacier boundary layer, so a clearer statement of assumptions and exploration of limitations is needed to trust the model outputs. Please contact me if you have any questions. I'm happy to provide the `.tex` file if it makes the response easier.

We thank you for the encouraging comments, especially for the fundamental questions on the underlying assumptions, and we will address the concerns below.

**Major comments**

> G1 A careful re-read for spelling and syntax throughout would improve the manuscript.

Thank you for the comment. We re-read the revised manuscript thoroughly and hope that the readability improved and no typos are present anymore.

> G2 Answering the second and third research questions hinges on WRF accurately resolving the stable boundary layer above the glacier surface. As stated in L324, this is a questionable assumption (although L55 suggests otherwise). As the authors found in Goger et al. (2022), the sensible heat flux was overestimated by roughly 2×. Although their grid is more coarse, Draeger et al. (2024) found an underestimation of sensible heat fluxes,

but also found that the results varied significantly depending on the model parameterization scheme chosen. As accurate modelling of the glacier boundary layer is paramount to the results presented, the manuscript would benefit greatly from a systematic enumeration of assumptions and limitations, and a clear argument of why the model outputs are to be trusted. Many of my specific comments throughout are related to this point.

- Thank you for this remark, which is also in accordance with referee #1's questions. We think we did not accurately emphasize that in our case study, the katabatic glacier boundary layer is either heavily disturbed or completely eroded, leading to neutral (REF, NO_UP) or even convective (NO_GL) vertical profiles. (cf. Fig. 4a-d in the manuscript). Therefore, we assume that on this case study day, larger-scale processes, i.e., the gravity wave, dominate over the local glacier boundary layer and ultimately erode it. The current LES setup is able to represent the relevant large-scale dynamical processes, while we agree, that small-scale glacier boundary layers are very likely not resolved appropriately on the grid. We would also like to mention at this point that we could very likely not use our model setup to investigate katabatically-driven flow regimes, because our model's grid spacing is too coarse and because underlying assumptions (e.g., MOST) break down in katabatically-driven situations. We added a paragraph on the model's limitations in Section 2:
  *"At the current horizontal grid spacing ($\Delta x$=48 m) not all scales in the LES are resolved equally well. As for all real-case LES, the scale separation problem (isolating the smaller from the larger scales) is inherent (Schemann et al., 2020). This leads to a 'better' representation of the larger scales (e.g., the dynamically-induced gravity wave), which is also evident in the NW day simulation of Goger et al. (2022), where they noted a too strong erosion of the glacier boundary layer by the gravity wave. Furthermore, we cannot expect that the small-scale stable boundary layer over HEF is resolved accordingly (both in the vertical and horizontal), because according to Cuxart (2015), a horizontal grid spacing of less than 10 m is necessary to simulate stable boundary layers in a realistic way. Still, since we focus in this study mostly on dynamically-driven processes, we think that we can provide important information on the impact of gravity wave formation on the glacier boundary layer flow development."*

- Indeed, the sensible heat flux is overestimated in REF (Goger et al., 2022), and we argued that this is related to the generally overestimated horizontal wind speeds in the model, a common bias in real-case LES over complex terrain with the WRF model (also found in Gerber et al., 2018; Umek et al., 2021; Liu et al., 2020). Despite this wind speed overestimation, a model validation with eddy-covariance observations showed other parameters like non-stationarity of sensible heat fluxes, the relation between sensible heat flux and wind speed, horizontal advection patterns, and turbulence kinetic energy were simulated in a realistic way. Therefore, we concluded that the underlying relevant physics are represented well in the 'NW day' simulations, the base for the current study.

- Draeger et al. (2024) ran their simulations on a coarser grid requiring a complete paramterization of the atmospheric boundary layer. Boundary layer paramterizations include simplified assumptions (e.g., assuming horizontally homogeneous and flat terrain, or excuding 3D effects; Goger et al., 2018, 2019), and these assumptions are clearly violated over complex terrain. To overcome these problems, we ran our simulations on a grid of $\Delta x = 48$ m, allowing us to use a LES closure instead to avoid the shortcomings of boundary-layer paramterizations. The Deardorff (1980) LES closure has the underlying assumption that the largest turbulent eddies are already resolved on the model grid. While this is not the case for small-scale stable boundary layers Cuxart (2015), we can assume that this criterion is met in our case study with $\Delta x = 48$ m where large-scale flows dominate. We would like to refer the referee to the detailed discussion in Goger et al. (2022), where we discuss all the aforementioned issues.

G3 Language like "deglaciation" or "removal of ice surfaces" is used to explain the case studies. How is this done? Is it a removal or a replacement? If the former, more explanation of how the underlying topography is inferred would be needed. If the latter, I would suggest changing the terminology and stating all of the ways in which this affects the WRF simulation (roughness lengths, temperature boundary condition, etc.).

- We agree that we have to be more consistent with our choice of words. Since "deglaciation" usually refers to glacier melting under a changing climate, and since our set-up does not follow realistic future glacier melting patterns, we decided to replace "degaliation" with "removing ice surfaces" in the revised manuscript.

- From the technical viewpoint, the model topography does not change by removing the glaciers in the domain. While in realtiy, melting glaciers lead to surface height change even within days (Voordendag et al., 2023), these surface height changes are not present in the current topography dataset we use (USGS, 2000).

| Land-use category | Albedo (%) | Moisture availability (%) | Emissivity (% at $9\,\mu$m) | $z_0$ (cm) | Thermal inertia (1 W m$^{-2}$ k$^{-1}$ s$^{1/2}$) |
|---|---|---|---|---|---|
| Snow or ice | 41.5 | 95 | 96.1 | 5 | 418 |
| Bare rock | 16.9 | 2 | 96.5 | 10 | 2948 |

Table 1: Surface parameters from the CORINE dataset for the two land-use categories 'snow or ice' and 'bare rock' for the summer season after Pineda et al. (2004).

- The only major change we implement by 'removing' the ice surfaces is changing the 'land use category' in the model input data from the category 'ice and snow' to 'bare rock'. According to Pineda et al. (2004), this leads to a changes in albedo, moisture availability, emissivity, roughness length, and thermal inertia (Tab. 1). We added the table together with an extended description on the changes in the senitivity studies with the changes in surface parameters to the manuscript, also in accordance with a request by Michael Haugeneder (referee #1).

G4 How does the "No Glaciers" experiment answers the questions asked? I think many of these results could go into the supplementary material, as they did not vary substantially from the "HEF-only" simulations (except on HEF). Might it be more illustrative to run the simulations with the upstream glaciers while removing HEF? This would also better simulate "realistic" deglaciation.

- We agree that the NO_UP simulation already answers most of our research questions. However, we conducted the NO_GL simulation to investigate, on the one hand, how the flow dynamics behave if entire catchment were ice-free, and on the other hand, to investigate the sole impact on HEF tongue on the flow structure. In our opinion, it is also an important result that the flow structure in the NO_GL simulation is very similar to NO_UP. One result from the NO_GL simulation is actually that HEF only (as in the NO_UP simulation), has a completely negligible impact on the general flow structure and the gravity wave development and breaking pattern, despite the negative heat fluxes (see heat flux figure of NO_UP). We actually think this is a major finding - namely, that under certain flow conditions, single glacier tongues are not able to maintain their own small-scale microclimate.

- Unfortunately, our project 'SCHISM' (Austrian Science Foundation, project number I 3841-N32, `https://dx.doi.org/10.55776/I3841`) and the associated computational resources already ended, therefore we cannot conduct additional simulations. However, since our study showed that the removal of the upstream glaciers has an impact on the stability of the upstream vertical profiles and therefore gravity wave formation, we think that keeping the upstream glaciers would lead to a gravity wave of a similar strength and structure as in the REF simulation.

- A follow-up project "Glacier-Space: Assessing the resilience and vulnerability of mountain ice masses" was recently funded by the Austrian Science Fund and some of the open questions arising from this manuscript can be answered in the future within the scope of this new project. We refer to future work on this in our conclusion section.

G5 Gravity wave "formation" is used throughout, including the title. Is gravity wave break-up not what is being simulated?

Both gravity wave formation (in the column above the upstream glaciers) and gravity wave breaking (in the column above HEF) are simulated in the model. The gravity wave forms over the upstream glaciers and breaks over the HEF valley.

G6 How representative is this 6h period of the typical flow conditions observed over HEF? This would help provide a take-away message that is more generalizable to other glaciers/study sites.

- Obleitner (1994) performed a wind climatology analysis for HEF and found that the katabatic glacier winds are mostly undisturbed by synoptic influence over the glacier tongue during summer. However, we have to

take into account that the glacier extent was larger than nowadays, and we could expect that HEF's katabatic glacier wind was more resilient to external forcing than nowadays (e.g., Shaw et al., 2024, showed with observations over multiple years that other flows from the thermally-induced valley wind circulation start to dominate over katabatic winds when ice surfaces are shrinking). At the South-facing slope facing HEF, Obleitner (1994) noted in their climatology a significant influence by Northerly gradient winds, which would correspond to a similar situation as our case study (Northerly/Northwesterly snyoptic flow).

- In a more recent dataset from the HEFEX campaign (summer 2018), Mott et al. (2020) found that around 20% of their wind data falls into the "disturbed" phase, i.e., an erosion of the katabatic glacier wind by lateral flows.

- We added the following paragraph to our discussion section:
*"A final open question to discuss is how representative our 6 hours of simulation are for HEF and its surroundings. Currently, we can only compare to a wind climatology at HEF compiled by Obleitner (1994), and they found a significant Northerly gradient wind influence on the South-facing slope of the valley, the same wind direction as in our case study. Furthermore, Mott et al. (2020) noted in the HEFEX campaign that in 20% of their wind observations the katabatic flow was 'disturbed' and the glacier boundary layer was eroded. Therefore, we can assume that the described situation of strong North-Westerly winds and gravity waves eroding the glacier boundary layer is not a single occurrence. Still, an updated wind climatology over HEF is necessary to quantify these events. Furthermore, applying an intermediate complexity model such as HICAR (Reynolds et al., 2023, 2024) to the region for entire seasons would shed more light on the typical wind patterns over HEF while being computationally cheaper than a full-physics LES."*

G7 The manuscript would benefit from a more holistic introduction to gravity waves. A schematic would make the interpretations of certain figures easier to follow. This should include a discussion of how WRF resolves gravity waves at the given scales. Stull (2017), p.761 highlights that the model grid spacing is not the model resolution, and wavelengths smaller than roughly $7\Delta x$ are often filtered out for numerical stability. What wavelengths do you expect to see, and how well does WRF resolve these wavelengths?

- We agree and re-organized our introduction. In the new version, and enitre section is dedicated to the formation and mechanisms of gravity waves.

- We only analyze the innermost domain of our simulations with a horizontal grid spacing of 48 m. To our knowledge, mostly sound waves are filtered out for stability in numerical models. Still, when we follow the $7\Delta x$ rule, this would mean that phenomena with a horizontal extent of 336 m can be resolved on our grid. Our entire glacier valley has a width of 3 km, and out innermost domain spans $12x12\,km^2$, corresponding to 251x251 grid points. Henceforth, we are confident that the gravity waves related to the flow over upstream topography is resolved on our grid.

- We added the following sentences in the discussion:
*"As discussed in Goger et al. (2022) and Voordendag et al. (2024), as our domain contains more than 10 grid points across the glacier valley, this suggests that the major ABL processes are resolved (Wagner et al., 2014). Furthermore, when we assume that effective horizontal resolution of the model is $7\Delta x$ (Stull, 2017, their pg. 761), phenomena of a horizontal extent of 336 m are resolved on the grid. This is the case for the gravity wave across the valley and the associated processes. Since we focus on the dynamical forcing and gravity waves, which form on scales larger than the small-scale glacier boundary layer, we can expect that the model delivers reliable results on these processes."*

G8 I find it hard to substantiate the conclusion that one must, in general, view local flow dynamics in the context of a larger system given that this experiment was only 6 h (especially without knowing how prevalent strong NW synoptic winds are at this location). While I know these simulations are expensive, I feel it would be illustrative to run a no-upstream-glaciers simulation under weaker, or otherwise different, synoptic forcing if this is the intended message.

- We agree that a single case study is possibly not representative for entire seasons or other glaciers/icefields. However, as outlined in our response to comment G6, Northerly winds are noticeable in wind climatology

of HEF and 20% of data collected during HEFEX (Mott et al., 2020) showed lateral disturbances of the katabatic wind. A possible additional analysis in the future could be exploring data from regional kilometric climate simulations (Molina et al., 2024) to identify situations with North-Westerly flow in the HEF region and quantify their frequency. However, this is out of scope of the current manuscript.

- Unfortunately, our project 'SCHISM' (Austrian Science Foundation, project number I 3841-N32, `https://dx.doi.org/10.55776/I3841`) and the associated computational resources already ended, therefore we cannot simulate another experiment with different or no synoptic forcing. However, we think that under weak synoptic forcing, local thermally-induced flows likely dominate, and they will interact with the katabatic down-glacier winds, similar to the situation in Shaw et al. (2024). The influence by the upstream glaciers would be smaller, and a similar situation as in Goger et al. (2022)'s "SW day" and as in the 'undisturbed' katabatic conditions in Mott et al. (2020) would be present. However, a follow-up project (Glacier Space, see answer above) was recently funded and is currently starting, so some of the open questions arising from this manuscript can be answered in the future.

> **G9** Can you comment more on the cause of the deviations between simulations and observations after 12:00? How do you argue that your experiment simulations are still valid at 12:00? In Fig 7, 8, and 10, the most pronounced changes (apart from the missing ice surfaces) seem to occur at the bottom of the domain, far from the removed ice surfaces. Is this physical? How far into the next valley do we see these effects?

- We draw our conclusions on the reliability of the REF simulation from the model validation with observations in Goger et al. (2022). Therefore, we use the REF simulation as a performance baseline for the two sensitivity runs, and since the simulation data of REF deviates from observed quantities (e.g., wind speed, temperature, etc, Fig. 2 in Goger et al., 2022), we do not analyze the results after 12:00 UTC.

- The correct simulation of mountain boundary layers is a major challenge for numerical models (Chow et al., 2019), and even increasing the horizontal resolution does not automatically improve results (Goger and Dipankar, 2024). One of the largest challenges for numerical models are so-called scale interactions between larger and smaller scales. While this problem might be less pronounced in a simulation with no synoptic forcing, in our case study, we have a strong synoptic, large-scale flow, interacting with smaller-scale processes in the glacier valley of HEF. Even for LES grid spacings, the scale interactions are a major challenge (e.g., foehn-cold air pool interactions or the interaction between slope flows and up-valley winds Umek et al., 2021; Goger and Dipankar, 2024). Furthermore, separating the larger and smaller scales (scale separation) is not possible (Schemann et al., 2020), evident in our simulations with the dominating large-scale flow. All the aforementioned issues accumulate with increasing simulation time, and in our case, this point is reached at 12:00 UTC, and this is case study specific. For example, in the winter case study with the same model setup, the WRF model is able to produce reliable results for a longer time period (Voordendag et al., 2024).

- Our focus of this manuscript was on HEF, the upstream glaciers, and surroundings. Still, we think the differences in the next valley are likely due to flow modification due to the missing ice surfaces and different gravity wave breaking patterns. Since the differences are in the range of less than 2 K, we deem the results as physical.

> **G10** I would prefer all equations to be included in the methods section. In general, I would encourage a slight restructuring so that the results presented can be better anticipated from the outset.

We added a new subsection called "analyses performed" to Section 2 where we present our analysis methods and the relevant equations.

**Specific comments**

> - I find that NO_UP and NO_GL read more like filenames than experiments, likely due to the underscore.

We agree to some extent, but we also think that the names of the sensitivity experiments should be distinguishable from the rest of the text, therefore we keep the underscores in the names. Furthermore, it is not uncommon to use underscores for numerical experiments (e.g., see Wagner et al., 2015, their Tab. 1).

> • L13: I do not think this study shows that a glacier tongue is never isolated from the surrounding glacier environment. (G8)

We re-wrote the sentence to
*"a single glacier tongue is not isolated from its environment under strong synoptic forcing [...]"*

> • L55: Can you argue this point more clearly? The scale difference between mesoscale and glacier-scale flow is at least a couple orders of magnitude. (G2)

We re-wrote the sentence to
*"With a horizontal mesh size of 48 m, the models can simulate mesoscale wind patterns on the glacier for both summer and winter successfully, but struggle wuith the correct representation fo small-scale glacier boundary layer features (Goger et al., 2022; Voordendag et al., 2024)."*

> • L105: Would you use a different module if the BL was not turbulent?

If turbulence were entirely parameterized (e.g., in a mesoscale simulation with kilometric grid spacing), the fluctuations in the simulation are reduced and henceforth no time averaging would be necessary.

> • L107: Three hours of spin-up time seems relatively low. Draeger et al. (2024) had a 24h spin-up time, and Liu et al. (2023) showed that the choice of spin-up time depends on process being modelled. To that end, Sun et al. (2014) highlights the importance of spin-up time in convective models. How are you certain that three hours is sufficient in the simulations without the stabilizing glacier surfaces? (G9)

Thank you for the comment, selecting the correct spin-up time is important for reliable LES simulations. As you correctly pointed out, the spin-up time for model depends on the physical phenomenon being modelled. For example, for convective events, the spin-up time is longer and might even need a correct/extended soil moisture initialization to deliver correct results.
The question on the spinup time of our model was also raised by referees for Goger et al. (2022). At that time we performed additional simulations with an earlier initialization, but noticed the same deterioration of model performance after around nine hours of simulation time, as in the current manuscript.
In our case, we simulate gravity waves and local mountain boundary layer processes over glaciers. Pfister et al. (2024) give an overview of the time scales of several phenomena in complex terrain (see Fig. R1). Since the timescales of local flow structures and exchange processes happen at timescales of less than an hour, we know that our spinup time of 3 hours is sufficient. We added an additional sentence to the model description:
*"The first three hours of simulation time are considered as spinup, to ensure that turbulence develops accordingly and given the time scale (an hour or less) given our phenomena of interest (gravity waves, boundary-layer processes)."*

> • L113 (and elsewhere): "Sensitivity study" reads more like testing different parameter regimes than removing entire glaciers.

The term "sensitivity study" is a widely used term for changing land surface or terrain properties in numerical models. Strictly speaking, removing the ice surfaces modifies the flow *regime* in our simulations. Furthermore, e.g. Umek and Gohm (2016) use the term for describing their numerical runs, where water bodies and/or topography are removed to study the mechanisms behind snowfall patterns.

> • L130: Non-stationarity meaning changing gradually throughout the day? Or referring to turbulent stationarity? If the latter, this also needs to be mentioned when introducing M-O theory.

We calculated the so-called 'non-stationarity ratio' by Mahrt (1998) in Goger et al. (2022) (their Fig. 8) from high-frequency time series of the sensible heat flux (both for observations and model output). A high non-stationarity ratio corresponds to large fluctuations of the sensible heat flux within the averaging period (in our case, 15 minutes), and non-stationarity was highest during the gravity wave breaking episodes, suggesting strong turbulence. We added to the brief summary of the phenomena in the REF simulation
*"The case study day was dominated by cross-glacier flow and high values of non-stationarity (Mahrt, 1998) of the sensible heat flux during gravity-wave breaking episodes and a strong mesoscale influence on the glacier boundary*

[Figure]

**Figure 17:** Schematic space-time diagram of resolvable scales during TEAMx-PC22 divided into (a) horizontally and (b) vertically resolvable scales. The Inn Valley target area (IVTA) and corresponding sub-target areas are indicated by colours. The filling of areas categorizes if the resolvable scale is inferred, partial, or continuous (cf. Section 4). A selection of processes and motions under investigation during TEAMx is also added (bold circles and text) to illustrate which of them were actually resolvable and at which location. All resolvable scales have a short description of which instruments were used (AWS: automatic weather station; lidar: remote sensing with light detection and ranging; UAS: uncrewed aerial system; DTS: distributed temperature sensing; radiosonde) and which parameters were retrieved (T: temperature; RH: relative humidity; p: pressure; u, v: horizontal wind speed and direction; w: vertical wind speed).

Figure R1: Figure from Pfister et al. (2024), their Fig. 17.

*layer."*

> • Figure 2 (and elsewhere): Here you present time as rows, and in other cases you present time as columns. Choosing one would be preferable. Additionally, axis labels here (and elsewhere) overlap.

We changed the overlap of labels (and in all further occurrences as well) and unified the row/column structure in all figures.

> • L157: Δ TKE?

Changed accordingly.

> • L163: Do you mean "shooting" downslope flow as classified by Mahrt (1982)? If so, please clarify this interpretation.

Thank you for the reference. However, the downslope flow in our simulations is not a gravity flow (like katabatic), but is rather dynamically forced due to large scale forcing, and displays hydraulic-jump-like features. We changed the formation to *"a distinct dynamically forced downslope flow"*.

> • Figure 4 a-d (and discussion): If the lowest level shown here is 3 m above the surface, then I am rather surprised that REF and NO_GL show a temperature difference of only 2 K. How do you explain the increase in temperature toward the surface in the lowest levels of REF? What does "mostly neutral" mean? (G2)

We agree that the difference in potential temperature in the lowest levels over HEF tongue is surprising. However, we have to keep in mind that the glacier tongue is under the influence of the cross-glacier flow, and any (local) glacier boundary layer (e.g., down-glacier katabatic winds) could not establish because of the strong lateral disturbance. Given that the mesoscale flow structure (and the resulting gravity wave inducing severe turbulence) in all three

simulations is similar, the weak temperature contrasts between REF, NO_GL and NO_UP are realistic.

> • Figure 4 e-h (and discussion): These subplots are visually dense. Some of these profiles look quite surprising
> – Are there two velocity maxima in the NO_ UP and NO_ GL simulations? What is meant by L198 "chaotic
> behaviour"? (G2)

We agree and changed the opacity of the wind direction points to increase the figure's readability.
As outlined in the sentence, due to the severe turbulence induced by the gravity wave, there is no distinguishable
pattern visible in the vertical profile, therefore we kept the term "chaotic behavior".

> • Figure 4 i-l (and discussion): I do not see how the Scorer parameter profiles presented show conditions
> favourable for gravity waves. In the provided references (e.g. Parmhed et al., 2004), the Scorer parameter is
> larger near the surface and then decreases, and is more than an order of magnitude larger than the near-surface
> values presented here. The Scorer profiles here look like they're being significantly affected by the division by
> a small U. Additionally, how are these gradients calculated? (G2)

We agree that we used a confusing strategy to explain the Scorer parameter and use the parameter in a different
way than Parmhed et al. (2004).
In the classical use of the Scorer parameter (see, e.g., this explanation, `https://resources.eumetrain.org/data/`
`4/452/navmenu.php?tab=4&page=5.0.0`), is to check whether conditions are favorable for gravity wave develop-
ment. The Scorer parameter is usually calculated in the vertical profiles (in our case, by using $\Delta z$ from the model's
grid) upstream of a mountain range or at the peak, as we do by calculating the Scorer parameter for the location
"upstream". In these vertical profiles, it is visible that the Scorer parameter has a sharp decrease in the REF sim-
ulation, but this sharp decrease is less distinct – or not existent – in NO_UP of NO_GL. Therefore, we conclude
that the upstream glaciers impose favorable conditions for gravity wave formation.
We update the text in the manuscript and removed the references (Parmhed et al., 2004; Söderberg and Parmhed,
2006), because we do not observe/simulate a katabatic flow in our time period of interest anyway.

> • L156-160: This presentation is a bit hard to follow. I would argue that the bulk method is agnostic to wind
> direction. That is, one does not check the wind direction to pick the temperature to use in the model, but
> rather uses whatever temperature is measured (which is likely different under different flow regimes, but then
> this is an implicit dependence, not an explicit one). Moreover, I'm a bit unsure why the bulk methods are
> being introduced here.

We re-wrote the paragraph to make the presentation more accessible. We present the bulk method here because
this is the method how surface fluxes are calculated in the model.

> • Equation 1: Brackets aren't tall enough.

Changed.

> • L174: $N = N(z)$ is correct, but a bit misleading as it hides the $\frac{\partial \theta}{\partial z}$ dependence.

We added the equation for $N(z)$ to the manuscript.

> • Figure 5: I think units of m would be preferable along the slope.

True, but our cross-section consists of several appended cross sections (see Fig. 1), so we thought keeping lat,lon is
better for the readability of the figure.

> • Figure 6: How do A and B relate to each other? It appears from A that the REF observations have the
> same variability (-0.5 to 1) as the NO_ UP and NO_ GL. Yet in B, The REF observations have a reduced
> range of observed wind directions relative to NO_ UP and NO_ GL.

Thank you for this remark. Indeed, in the REF simulation, almost all blue points are on the diagonal and
correspond to the dynamical regime. Some of these blue points indeed collapse over each other on the diagonal and
are therefore 'on top of each other', unfortunately.
To answer the second remark, the range of wind direction is reduced because the up-valley flow only establishes

much later in the REF simulations (because of the still stronger gravity wave), while in the NO_UP and NO_GL simulations, the yup-valley flow establishes much ealier and increased turbulence over the glacier surface (compared to REF) also leads to a wider range and chaotic structure of the wind directions.

Furthermore, we double-checked the python script behind the figure again. All three simulations use the same number of data points, but the blue data points (especially in the lower right part of the plot) collapse on each other in the dynamical regime, and/or are concealed by the points from the NO_UP and NO_GL simulations (esp. in the case of the channelling regime).

> • Equation 2: Typeset this equation using "upright" cos and brackets of appropriate height. I also prefer UWI and wdir to be upright and not slanted here and in the text. The conversion to radians is implied. So I prefer,
>
> $$\mathrm{UWI} = \cos |\mathrm{wdir} \quad \phi|,$$
>
> or similar. That said, if the UWI is only computed at one point along the glacier, I feel that wind direction alone (perhaps oriented such that 0 is upglacier) is sufficient, and simpler. It seems the utility of this metric arises when comparing observations from multiple locations where "upslope" might be different.

Thank you, we changed cos to "upright", but not the rest of the equation, since Weather and Climate Dynamics' guidelines require italics for variables in equations (see `https://www.weather-climate-dynamics.net/submission.html`, "Mathematical notation and terminology").
We prefer to keep Shaw et al. (2023)'s proposed formula to calculate UWI, because it also takes glacier/valley orientation into context. This formula has the potential to be widely used in studying wind regimes over glaciers in the future because it only requires few observations, and we would want to demonstrate its usability/potential.

> • L223: Worthwhile to introduce the negative case. Perhaps more clear to state, say, |UWI| < 0.5 indicates cross-glacier flow.

Added to the manuscript:
"When $UWI = 1$, the flow is exactly up-glacier, while $UWI \approx 0.5$ indicates cross-glacier flow, and $UWI \approx 0$ implies down-glacier flow."

> • L234: Why use upstream location and not wind direction from a higher level in the same column?

We prefer to use the upstream location instead of the column directly above HEF tongue, because our focus of the manuscript is on the change in upstream conditions (removing glaciers) and their impact on the flow structure. Another reason is that measurements with a weather station are potentially possible as in (Whiteman and Doran, 1993), in contrast to the location directly above a glacier, where a vertical profiling method would be necessary. Furthermore, we want to stay consistent with our location selection with Fig. 4.

> • Section 3.3: The latter half reads as a discussion and not a presentation of results.

We agree, but since we use Section 3.3 to explain the observed patterns in the previous sections, we think that it is appropriate to discuss the results at this point.

> • L276-281: As you say, none of these results are particularly surprising. (G4)

Yes. We moved Figure 8 to the appendix.

> • Figures 7 and 8: The choice of colorbar scales makes interpretation challenging. I would be far more interested to see the differences in sensible heat flux into the glaciers. A colorbar with a nonlinear scaling might help here. Overlapping axis labels. Inconsistent sensible heat flux vs. "SH" (here and in text).

Thank you, we removed the sensible heat fluxes of the surroundings and now focus on the (missing) ice surfaces only.

- Axis labels do not overlap anymore.

- We removed the "SH" abbreviations (both from figure and text).

• Section 4.2: Please clarify the intended message of this section. The beginning sentences do not seem related to the section title, directly. It seems the focus is on advection of heat and not momentum?

We agree and added introductory sentences at the beginning of the section:
*"Observations and numerical simulations agree that the local flow patterns over HEF strongly affect the heat transport and advection processes over the glacier tongue (Mott et al., 2020; Goger et al., 2022; Haugeneder et al., 2024). Therefore, we explore the impact of changing the ice surfaces in the next paragraphs."*

• L290-292: What would be the foundation of this assumption?

This idea was proposed by Mott et al. (2020), but was, however, disproven by identifying the gravity wave being the major mechanism between advection patterns on HEF tongue (Goger et al., 2022).

• Equation 3: This is not a complete heat budget. Please explain which terms are omitted and why. What length scales are the derivatives taken over? ADV and vHFD appear upright in the text, so should appear upright here. That said, they are only used this once so don't need to be abbreviated.

We added an additional sentence on our assumptions:
*"We neglect the radiative flux divergence as in (Goger et al., 2022), because we consider it small during daytime. Horizontal averages are taken over $\Delta x=48\,m$."*
Furthermore, we removed the abbrevations ADV and vHFD.

• Figure 9: I prefer the colorbar only over the subpanels where it is relevant. (d-f) I feel would be better presented as components of the budget. Is the budget closed? I can't tell if this is the LHS or RHS of eqn. 3 at present. (c) Downglacier advection? Or total horizontal advection?

We split, also based on the suggestion from the other referee, into two separate figures. Panels (a)-(c) from old Fig. 8 are now new Figure 8, while the new Fig. 9 includes panels (d)-(f) from old Fig. 8, while adding a row of panels of the time-averaged components of the total vertical heat budget to the respective simulations. We constrained to the colorbar to panels (a)-(c). We also added extra text to describe the behaviour of the components of the vertical heat budget at HEF tongue.

• Figure 10: Similar to before, is the message here related to the temperature differences over the whole domain, or the temperature differences on the glacier? The colorbars could show this better, if the latter. Panel d is not needed.

We calculated time averages over our simulation period of interest (06:00-12:00 UTC) over the entire domain, as Fig. 9 shows, so the latter case is true. Furthermore, we indicate in the text description whether we are currently talking about the glacier or about close-by surroundings. We removed panel (d) in the Figure.

• L313: Why do you trust this assessment of 2 m temperature given the criticisms of M-O theory applied to katabatics (e.g., Grisogono et al., 2007, and references therein). (G2)

We are aware of the breakdown of MOST in katabatic flows over glaciers. However, as we state in the manuscript, an analysis of surface temperature output from the model would not make much sense (since it is constant at 0°C over the ice surface). During our time period of interest used for the averaging in Fig. 9, there is no katabatic glacier wind observed, since the glacier boundary layer is disturbed due to the cross-glacier flow (Mott et al., 2020; Goger et al., 2022). Therefore, we can trust the 2 m temperature values to some extent, if we keep in mind that MOST has several other shortcoming and needs substantial revision Stiperski and Calaf (2023), but unfortunately, this is the only option we currently have to compare the temperature patterns from model output over HEF. We added a clarifying sentence
*"However, although MOST breaks down in katabatic flows (Grisogono et al., 2007) and is in need of substantial revision (Stiperski and Calaf, 2023), we do not observe katabatic winds in our time period of interest, since the glacier boundary layer is heavily disturbed by the cross-glacier flow, as outlined in the previous sections and in Mott et al. (2020); Goger et al. (2022)."*

• L330: What are dynamical aspects?

We agree that this formulation is unclear. We changed it to "dynamical forcing".

- L346: Earlier what?

Changed to
*"breaks 30 minutes earlier"*

- L355: If this is an intended take-away message, it would be good to (1) make this a more clear objective, and (2) quantify these effects more clearly. How did you decide on 5 km? The Columbia Icefield is the study of Conway et al. (2021) is very large. Do you expect the size of the icefield to play a role? Do you expect this to be true in all synoptic conditions or just some? (G6/G8)

Thank you for the remark. We touch upon the spatial scale of our glacier catchment in the discussion, but we added more sentences discussing other icefields in the discussion as well:
*"Given that, it makes sense to speak of a system of glaciers influencing each other's local micro-climates on a scale of around 5-10 km, given the scale of HEF and its surroundings. However, this length scale for the development of mesoscale ice breezes - similar as in Conway et al. (2021) - depends on the size of the glaciers and connected upstream ice fields. An example for larger icefields influencing the local wind patterns on glacier tongue would be the Columbia icefield in Canada (Conway et al., 2021), the Jostedalsbreen icefield in Western Norway (Haualand et al., 2024), or Vatnajökull icecap in Iceland Björnsson et al. (2005). All of them are larger than 50 km, and dependent on flow direction, icefield breezes or gravity waves are favorable to develop. Our work allowed us to shed light on the impact of upstream icefields/glaciers on local glacier boundary layers, but more detailed research in combination with wind climatologies and flow-resolving simulations are necessary in the future."*

**References**

Björnsson, H., Gudmundsson, S., and Pálsson, F.: Glacier winds on Vatnajökull ice cap, Iceland, and their relation to temperatures of its lowland environs, Ann. Glaciol., 42, 291–296, https://doi.org/10.3189/172756405781812493, 2005.

Chow, F. K., Schär, C., Ban, N., Lundquist, K. A., Schlemmer, L., and Shi, X.: Crossing Multiple Gray Zones in the Transition from Mesoscale to Microscale Simulation over Complex Terrain, Atmosphere, 10, https://doi.org/10.3390/atmos10050274, 2019.

Conway, J. P., Helgason, W. D., Pomeroy, J. W., and Sicart, J. E.: Icefield Breezes: Mesoscale Diurnal Circulation in the Atmospheric Boundary Layer Over an Outlet of the Columbia Icefield, Canadian Rockies, J. Geophys. Res. Atmos., 126, e2020JD034 225, https://doi.org/10.1029/2020JD034225, 2021.

Cuxart, J.: When Can a High-Resolution Simulation Over Complex Terrain be Called LES?, Front. Earth Sci., 3, 6, https://doi.org/10.3389/feart.2015.00087, 2015.

Deardorff, J. W.: Stratocumulus-capped mixed layers derived from a three-dimensional model, Boundary-Layer Meteorol., 18, 495–527, https://doi.org/10.1007/BF00119502, 1980.

Draeger, C., Radić, V., White, R. H., and Tessema, M. A.: Evaluation of reanalysis data and dynamical downscaling for surface energy balance modeling at mountain glaciers in western Canada, The Cryosphere, 18, 17–42, https://doi.org/10.5194/tc-18-17-2024, 2024.

Gerber, F., Besic, N., Sharma, V., Mott, R., Daniels, M., Gabella, M., Berne, A., Germann, U., and Lehning, M.: Spatial variability in snow precipitation and accumulation in COSMO–WRF simulations and radar estimations over complex terrain, The Cryosphere, 12, 3137–3160, https://doi.org/10.5194/tc-12-3137-2018, 2018.

Goger, B. and Dipankar, A.: The impact of mesh size, turbulence parameterization, and land-surface-exchange scheme on simulations of the mountain boundary layer in the hectometric range, Q. J. R. Meteorol. Soc., 150, 3853–3873, https://doi.org/10.1002/qj.4799, 2024.

Goger, B., Rotach, M. W., Gohm, A., Fuhrer, O., Stiperski, I., and Holtslag, A. A. M.: The Impact of Three-Dimensional Effects on the Simulation of Turbulence Kinetic Energy in a Major Alpine Valley, Boundary-Layer Meteorol, 168, 1–27, https://doi.org/10.1007/s10546-018-0341-y, 2018.

Goger, B., Rotach, M. W., Gohm, A., Stiperski, I., Fuhrer, O., and de Morsier, G.: A New Horizontal Length Scale for a Three-Dimensional Turbulence Parameterization in Mesoscale Atmospheric Modeling over Highly Complex Terrain, J. Appl. Meteor. Climatol., 58, 2087–2102, https://doi.org/10.1175/JAMC-D-18-0328.1, 2019.

Goger, B., Stiperski, I., Nicholson, L., and Sauter, T.: Large-eddy simulations of the atmospheric boundary layer over an Alpine glacier: Impact of synoptic flow direction and governing processes, Q. J. R. Meteorol. Soc, 148, 1319–1343, https://doi.org/10.1002/qj.4263, 2022.

Grisogono, B., Kraljević, L., and Jeričević, A.: The low-level katabatic jet height versus Monin–Obukhov height, Q. J. R. Meteorol. Soc., 133, 2133–2136, https://doi.org/10.1002/qj.190, 2007.

Haualand, K. F., Sauter, T., Abermann, J., de Villiers, S. D., Georgi, A., Goger, B., Dawson, I., Nerhus, S. D., Robson, B., Hauknes Sjursen, K., Thomas, D., Thomaser, M., and Yde, J. C.: Micro-Meteorological Impact of Glacier Retreat and Proglacial Lake Temperature in Western Norway, ESS Open Archive, https://doi.org/10.22541/essoar.172926901.19613096/v1, 2024.

Haugeneder, M., Lehning, M., Stiperski, I., Reynolds, D., and Mott, R.: Turbulence in the Strongly Heterogeneous Near-Surface Boundary Layer over Patchy Snow, Boundary-Layer Meteorol., 190, https://doi.org/10.1007/s10546-023-00856-4, 2024.

Liu, Y., Liu, Y., Muñoz Esparza, D., Hu, F., Yan, C., and Miao, S.: Simulation of Flow Fields in Complex Terrain with WRF-LES: Sensitivity Assessment of Different PBL Treatments, J. Appl. Meteor. Climatol., 59, 1481–1501, https://doi.org/10.1175/JAMC-D-19-0304.1, 2020.

Liu, Y., Zhuo, L., and Han, D.: Developing spin-up time framework for WRF extreme precipitation simulations, Journal of Hydrology, 620, 129 443, https://doi.org/10.1016/j.jhydrol.2023.129443, 2023.

Mahrt, L.: Momentum Balance of Gravity Flows, J Atmos Sci, 39, 2701 – 2711, https://doi.org/10.1175/1520-0469(1982)039<2701:MBOGF>2.0.CO;2, 1982.

Mahrt, L.: Flux Sampling Errors for Aircraft and Towers, J. Atmos. Ocean. Technol., 15, 416 – 429, https://doi.org/10.1175/1520-0426(1998)015<0416:FSEFAA>2.0.CO;2, 1998.

Molina, M. O., Careto, J. M., Gutiérrez, C., Sánchez, E., Goergen, K., Sobolowski, S., Coppola, E., Pichelli, E., Ban, N., Belušić, D., Short, C., Caillaud, C., Dobler, A., Hodnebrog, Ø., Kartsios, S., Lenderink, G., de Vries, H., Göktürk, O., Milovac, J., H, F., Truhetz, H Demory, M. E., Warrach-Sagi, K., Keuler, K., Adinolfi, M., Raffa, M., Tölle, M., Sieck, K., Bastin, S., and Soares, P. M. M.: The added value of simulated near-surface wind speed over the Alps from a km-scale multimodel ensemble, Clim. Dyn., 62, 4697–4715, https://doi.org/10.1007/s00382-024-07257-4, 2024.

Mott, R., Stiperski, I., and Nicholson, L.: Spatio-temporal flow variations driving heat exchange processes at a mountain glacier, The Cryosphere, 14, 4699–4718, https://doi.org/10.5194/tc-14-4699-2020, 2020.

Obleitner, F.: Climatological features of glacier and valley winds at the Hintereisferner (Ötztal Alps, Austria), Theor. Appl. Climatol., 49, 225–239, https://doi.org/10.1007/BF00867462, 1994.

Parmhed, O., Oerlemans, J., and Grisogono, B.: Describing surface fluxes in katabatic flow on Breidamerkurjökull, Iceland, Q. J. R. Meteorol. Soc., 130, 1137–1151, https://doi.org/10.1256/qj.03.52, 2004.

Pfister, L., Gohm, A., Kossmann, M., Wieser, A., Babić, N., Handwerker, J., Wildmann, N., Vogelmann, H., Baumann-Stanzer, K., Alexa, A., Lapo, K., Paunović, I., Leinweber, R., Sedlmeier, K., Lehner, M., Hieden, A., Speidel, J., Federer, M., and Rotach, M. W.: The TEAMx-PC22 Alpine field campaign - Objectives, instrumentation, and observed phenomena, Meteorol Z, 33, 199–228, https://doi.org/10.1127/metz/2024/1214, 2024.

Pineda, N. Jorba, O., Jorge, J., and Baldasano, J. M.: Using NOAA AVHRR and SPOT VGT data to estimate surface parameters: application to a mesoscale meteorological model, Int. J. Remote Sens., 25, 129–143, https://doi.org/10.1080/0143116031000115201, 2004.

Reynolds, D., Gutmann, E., Kruyt, B., Haugeneder, M., Jonas, T., Gerber, F., Lehning, M., and Mott, R.: The High-resolution Intermediate Complexity Atmospheric Research (HICAR v1.1) model enables fast dynamic downscaling to the hectometer scale, Geosci. Model Dev., 16, 5049–5068, https://doi.org/10.5194/gmd-16-5049-2023, 2023.

Reynolds, D., Quéno, L., Lehning, M., Jafari, M., Berg, J., Jonas, T., Haugeneder, M., and Mott, R.: Seasonal snow–atmosphere modeling: let's do it, The Cryosphere, 18, 4315–4333, https://doi.org/10.5194/tc-18-4315-2024, 2024.

Schemann, V., Ebell, K., Pospichal, B., Neggers, R., Moseley, C., and Stevens, B.: Linking Large-Eddy Simulations to Local Cloud Observations, J Adv Model Earth Sys, 12, e2020MS002 209, https://doi.org/10.1029/2020MS002209, e2020MS002209 10.1029/2020MS002209, 2020.

Shaw, T. E., Buri, P., McCarthy, M., Miles, E. S., Ayala, A., and Pellicciotti, F.: The Decaying Near-Surface Boundary Layer of a Retreating Alpine Glacier, Geophys Res Lett, 50, e2023GL103 043, https://doi.org/10.1029/2023GL103043, 2023.

Shaw, T. E., Buri, P., McCarthy, M., Miles, E. S., and Pellicciotti, F.: Local Controls on Near-Surface Glacier Cooling Under Warm Atmospheric Conditions, J Geophys Res. Atmospheres, 129, e2023JD040 214, https://doi.org/10.1029/2023JD040214, e2023JD040214 2023JD040214, 2024.

Söderberg, S. and Parmhed, O.: Numerical Modelling of Katabatic Flow Over a Melting Outflow Glacier, Boundary-Layer Meteorol., 120, 509–534, https://doi.org/10.1007/s10546-006-9059-3, 2006.

Stiperski, I. and Calaf, M.: Generalizing Monin-Obukhov Similarity Theory (1954) for Complex Atmospheric Turbulence, Phys. Rev. Lett., 130, 124 001, https://doi.org/10.1103/PhysRevLett.130.124001, 2023.

Stull, R. B.: Practical Meteorology: An Algebra-based Survey of Atmospheric Science, Univ. of British Columbia, URL https://www.eoas.ubc.ca/books/Practical_Meteorology/, 2017.

Sun, J., Xue, M., Wilson, J. W., Zawadzki, I., Ballard, S. P., Onvlee-Hooimeyer, J., Joe, P., Barker, D. M., Li, P.-W., Golding, B., Xu, M., and Pinto, J.: Use of NWP for Nowcasting Convective Precipitation: Recent Progress and Challenges, Bull. Amer. Meteorol. Soc., 95, 409 – 426, https://doi.org/10.1175/BAMS-D-11-00263.1, 2014.

Umek, L. and Gohm, A.: Lake and Orographic Effects on a Snowstorm at Lake Constance, Mon Wea Rev, 144, 4687 – 4707, https://doi.org/10.1175/MWR-D-16-0032.1, 2016.

Umek, L., Gohm, A., Haid, M., Ward, H. C., and Rotach, M. W.: Large eddy simulation of foehn-cold pool interactions in the Inn Valley during PIANO IOP2, Q. J. R. Meteor. Soc., 147, 944–982, https://doi.org/10.1002/qj.3954, 2021.

USGS: Shuttle Radar Topography Mission (SRTM) 1 Arc-SecondGlobal Dataset, https://doi.org/10.5066/F7PR7TFT, 2000.

Voordendag, A., Prinz, R., Schuster, L., and Kaser, G.: Brief communication: The Glacier Loss Day as an indicator of a record-breaking negative glacier mass balance in 2022, The Cryosphere, 17, 3661–3665, https://doi.org/10.5194/tc-17-3661-2023, 2023.

Voordendag, A., Goger, B., Prinz, R., Sauter, T., Mölg, T., Saigger, M., and Kaser, G.: A novel framework to investigate wind-driven snow redistribution over an Alpine glacier: combination of high-resolution terrestrial laser scans and large-eddy simulations, The Cryosphere, 18, 849–868, https://doi.org/10.5194/tc-18-849-2024, 2024.

Wagner, J. S., Gohm, A., and Rotach, M. W.: The Impact of Horizontal Model Grid Resolution on the Boundary Layer Structure over an Idealized Valley, Mon. Wea. Rev., 142, 3446–3465, https://doi.org/10.1175/MWR-D-14-00002.1, 2014.

Wagner, J. S., Gohm, A., and Rotach, M. W.: Influence of along-valley terrain heterogeneity on exchange processes over idealized valleys, Atmos. Chem. Phys., 15, 6589–6603, https://doi.org/10.5194/acp-15-6589-2015, 2015.

Whiteman, C. D. and Doran, J. C.: The Relationship between Overlying Synoptic-Scale Flows and Winds within a Valley, J. Appl. Meteor., 32, 1669–1682, https://doi.org/10.1175/1520-0450(1993)032<1669:TRBOSS>2.0.CO;2, 1993.

---

## Referee Report (RR1)

**Second Review of "Investigating the influence of changing ice surfaces on gravity wave formation impacting glacier boundary-layer flow with large-eddy simulations" by Goger et al.**

Michael Haugeneder, michael.haugeneder@slf.ch

December 9, 2024

The authors have sufficiently improved the manuscript. I just have some (last) minor comments.

**Minor Comments**

1. Figure 1: Could you enlarge the cross to enhance the visibility?

2. l. 127f.: Could you check the sentence structure?

3. Figure 3: Please check the panel attribution to the simulation cases in the caption.

4. Figure 4: x-axis label in i-l) should be $l^2$

5. Figure 5: Please check the panel attribution to the simulation cases in the caption.

6. Figure 6 caption: swap "black point" and "black cross"

7. l. 307: two times "the"

8. l. 308: "... REF simulation  compared to ..."

9. Figure 7: add dashed blue lines in NO_UP column

10. l. 313f. The reasoning here is still unclear to me. Why are the sensible heat fluxes strongly dependent on wind *speed as the surface temperature is* $0°$C? In the Monin–Obukhov formula, wind speed is just an factor equally important as the temperature difference and the stability correction, right? Do you mean wind *direction*?

11. l 316f. Figure 7b shows a positive difference between NO_UP and REF, which indicates *weaker* negative fluxes in NO_UP, right?

12. Figure 8: 12 : 00, j-l (not i)

13. Figure 9$a$: label right y-axis (wind direction)

14. l. 358f.: Isn't there both cooling and warming of the atmosphere above 2850m a.m.s.l. between 09 : 00 and 12 : 00?

15. Figure 10 right column: What does "net" refer to? Should it not be the sum of advective and vertical heat flux divergence contributions (as stated in eq. 5)?

16. l. 373: Fig. 11d instead of 11c?

17. l. 374: The authors refer to warm air advection shown in Fig. 9. But in Fig. 10b and e cold air advection is indicated in the profiles.

18. l. 377f.: Could the authors comment on the cooling pattern visible in Fig. 11d and e at the very bottom of the domain at higher elevations?